# ReNF: Rethinking the Design Space of Neural Long-Term Time Series Forecasters

## Abstract

Neural Forecasters (NFs) are a cornerstone of Long-term Time Series Forecasting (LTSF). However, progress has been hampered by an overemphasis on architectural complexity at the expense of fundamental forecasting principles. In this work, we return to first principles to redesign the LTSF paradigm. We begin by introducing a Multiple Neural Forecasting Proposition that provides a theoretical motivation for our approach. We propose Boosted Direct Output (BDO), a novel forecasting paradigm that synergistically hybridizes the causal nature of Auto-Regressive (AR) models with the stability of Direct Output (DO). In addition, we stabilize the learning process by smoothly tracking the model's parameters. Extensive experiments show that these principled improvements enable a simple MLP to achieve state-of-the-art performance, outperforming recent, complex models in nearly all cases, without any specific considerations in the area. Finally, we empirically verify our proposition, establishing a dynamic performance bound and identifying promising directions for future research. The code for review is available at: `https://anonymous.4open.science/r/ReNF-A151`.

## 1 Introduction

The progression of any real-world event is a unique, non-repeatable process, often governed by chaotic dynamics we cannot perfectly measure (Robertson, 1929) or describe. This implies that any observed time series is just one stochastic sample from a complex underlying system. Consequently, a fundamental open question persists: how can we best estimate the long-term future states depending only on a single, observed historical sequence?

Deep Neural Networks (DNNs) have recently received increasing attention in Long-Term Time Series Forecasting (LTSF) (Kim et al., 2025). Their ability to model high-dimensional, non-linear dependencies makes the Neural Forecaster (NF) a promising tool for capturing complex temporal dynamics and implicit dependencies (Laiz et al., 2024). However, the literature reveals several critical problems that should be addressed to unlock the full potential of DNNs in this domain.

One confusing puzzle is that both advanced architectures, like transformer-based (Nie et al., 2023) and simple linear-based models (Zeng et al., 2023), are reported to achieve state-of-the-art performance interchangeably while their complexity varies significantly. This is partly because some datasets prefer parsimonious (Deng et al., 2024) while the others have sufficient volume for complex models to fit. But perhaps a deeper issue is the insufficient exploration of the networks' intrinsic capabilities. As indicated by (Lu et al., 2025), many NFs contain redundant components and are used selectively depending on the property of the data, resulting in their full potential remaining untapped without extensive tuning or regularizations.

Furthermore, the field's focus has shifted towards designing specialized modules for properties that are believed to be beneficial for LTSF, such as multi-scale (Wang et al., 2024b) and non-stationary (Liu et al., 2022). However, the progress has become erratic because these architectural additions often yield subtle gains while overlooking fundamental principles. We re-emphasize that progress in this field becomes circuitous when we introduce advanced models merely according to the empirical conclusions from other realms, but overlook the detailed instructions for their full utilization. A primary and more direct path to advancement may lie in fundamentally improving the training stability and generalization capabilities of NFs themselves.

Beyond model architecture, we find that the dominant Direct Output (DO) framework does not make full use of the available supervision. In a standard DO setup, an NF is trained to predict the entire future horizon in a single forward pass, meaning the label information is leveraged only once per optimization step (see Sec.2.3 for the detailed explanation). This encourages the model to finally learn a monolithic representation that maps the entire history to the entire future, without explicitly modeling the sequential dependencies within the forecast itself. We contend that this approach hinders the model from developing a more granular, causal understanding of the future, thereby limiting its full potential.

To address these fundamental shortcomings, this paper proposes a new LTSF framework, derived from first principles, that establishes a reliable and high-performing benchmark model. Our contributions are summarized as follows:

- We present a Multiple Neural Forecasting Proposition (MNFP) with empirical evaluations, which provides a formal motivation for employing multiple Neural Forecasters (NFs) in Long-Term Time Series Forecasting (LTSF).

- We redesign the forecasting architecture by introducing a novel, streamlined forecasting paradigm that synergistically combines the strengths of Direct Output (DO) and Auto Regressive (AR) methods. Furthermore, we apply the Exponential Moving Average (EMA) technique to effectively stabilize the convergence of Neural Forecasters.

- We conduct extensive experiments to demonstrate that a pure MLP-based forecaster, when trained with our paradigm, can significantly outperform recent state-of-the-art models on nearly all standard LTSF benchmarks. These results establish a new and efficient performance baseline for the field.

## 2 METHOD

In this section, we lay out the theoretical and methodological foundations of our work. We begin by introducing a core proposition that provides the theoretical underpinnings for our approach. To build intuition, we then illustrate our designs using a simple MLP architecture on carefully selected datasets that highlight the key effects of our methods. The comprehensive experimental setup and full results on benchmark datasets are detailed in Sec.3 and the Appendix.

### 2.1 PRELIMINARIES

**Neural Forecasting Machine (NFM).** In this work, we denote the NFM as a forecaster modeled by neural nets and satisfying that, given an input time series $X \in \mathbb{R}^{t_x}$, an NFM yields one and only one series $Y \in \mathbb{R}^{t_y}$ through the operation $Y = \text{NFM}[t_x, t_y, \theta, \gamma](X)$, where $t_x$ and $t_y$ denote the input and output lengths. The symbol $\theta$ denotes the model's parameters, and $\gamma$ denotes the random state encompassing all random factors such as the computational environment and exogenous variables.

**Forecasting Task.** We characterize a time series forecasting task with four parameters $X_h, X_f, \hat{Y}_f, Y_f$, where $X_h, X_f$ denote the history and future portions of an *observed time series*. $Y_f$ denotes the real unobserved future data, while $\hat{Y}_f$ is an empirical forecast generated by an NFM with input $X_f$. Since we treat the observed data $X$ as a noisy sample of the true underlying signal, the ultimate target is to learn a mapping from the history input $X_h$ to a prediction $\hat{Y}_f$ that best approximates the $Y_f$.

### 2.2 THEORETICAL MOTIVATION FOR MULTIPLE FORECASTS

The following proposition provides the theoretical motivation for our work. For simplicity, we present the proof for the univariate case; however, the proposition can be readily extended to the multivariate setting.

**Proposition 1** (Multiple Neural Forecasting Proposition (MNFP)). *In the context of standard machine learning. Given a NFM $\Phi(t_x, t_y, \theta, \gamma)$ and an observed time series $X_h = (x_1, x_2, \cdots, x_n)$ where each element $x_t$ is drawn from a true distribution $p_t(\mu_t, \sigma_t^2)$ with mean $\mu$ and standard deviation $\sigma_t$. One can generate a series by $\Phi$: $\hat{Y}_f = (y_1, y_2, \cdots, y_T)$ with $y_t$ drawn from the expected forecast*

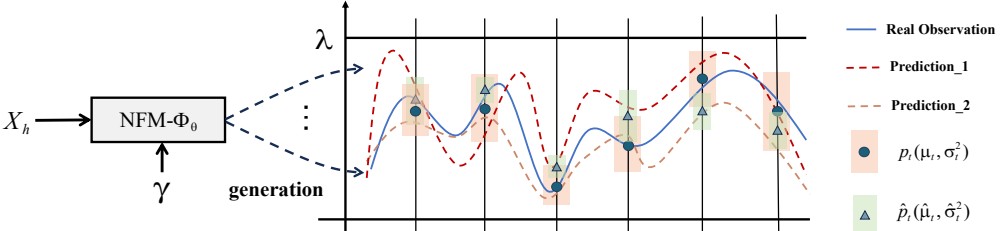

Figure 1: A single trained NFM $\Phi_\theta$ can generate multiple forecasts from a fixed input $X_h$ under various states $\gamma$. These forecasts are expected to follow the empirical process $(\hat{p}_t, \ t = 1, 2, \cdots)$ approximated by the NFM with the observed data. The model-expected process is distinguished from the real data distribution $(p_t, \ t = 1, 2, \cdots)$ by a bias.

distribution $\hat{p}(\hat{\mu}_t, \hat{\sigma}_t^2)$ such that $\hat{Y}_f$ approaches $Y_f$ almost surely, and the error between $\hat{Y}_f$ and $X_f$ is bounded.

Specifically, a trivial but informative upper bound on the $l_1$ error is given by $T(\lambda + \sqrt{c}(b + \sigma_t))/\sqrt{c}$ with the following parameters/conditions:

- *(A1) The objective time series with finite length is bounded by a positive real constant $\lambda$.*

- *(A2) The NFM generates candidate series $\{\hat{Y}_f^{(i)}\}_{i=1}^{c>1}$ under various $\theta^i$ or random states $\gamma^i$.*

- *(A3) The predictive bias of the NFM is bounded by up to $\sup_t |\hat{\mu}_t - \mu_t| = b$.*

The proof and further analysis can be found in Appendix C. To interpret, the shown bound value provides us with some intuitions: for instance, the forecasting error of the NFM naturally grows with the forecast horizon $T$ and the data range $\lambda$, because of the expanded solution space. We emphasize that condition A3 distinguishes between the observed real series and the expected true series. This frames the observed data not as the absolute ground truth, but as a single, near-optimal sample from the underlying stochastic process. While it is feasible to theoretically approach the expected forecast, the empirical gap between any observation $x_t$ and the true expectation $\mu_t$ is practically irreducible.

The most critical insight of the proposition is that an accurate forecast is theoretically approachable even with a rather weak generator, provided a sufficiently large number of candidate outputs is produced. This proposition is rooted in the well-established theory of ensemble methods (Bates & Granger, 1969), which demonstrates that combining multiple diverse estimators can lead to a more accurate and robust final estimate. Furthermore, higher-quality historical data and a more powerful NFM (which reduces the estimation error $b$) both serve to constrain the solution space, leading to better predictions. Inspired by this, we focus on reducing the total error variance by increasing $c$, which has been largely overlooked in the recent pursuit of complex, monolithic architectures.

**Post combination of multiple forecasts.** Given a set of candidate outputs from an NFM, there exist combinatorial methods to synthesize a single, integrated forecast that would be more accurate than any individual candidate. We can represent such a method as an unknown functional $g_c$ that produces the post-combined forecast as:

$$\hat{Y}_{pc} = g_c(\{\hat{Y}_f^{(i)}\}_{i=1}^c) \tag{1}$$

In our experiments, we will empirically show the potential of this combination on real-world datasets. In practice, however, directly creating an optimal combination function without access to future information is a significant challenge (Clemen, 1989). Therefore, designing a framework that can implicitly and effectively leverage these multiple forecasts is one of the key principles guiding our subsequent work.

### 2.3 NOVEL FORECASTING PARADIGM

The current frameworks in LTSF involve two main strategies: 1-step Auto Regressive (AR(1)) and Direct Output (DO). While AR models perform consistently with recurrent state space models like RNNs (Siami-Namini et al., 2019), they are known

Figure 2: Features of DO and BDO.

to suffer from significant error accumulation in long-horizon forecasting and are empirically evidenced to largely fall behind DO in LTSF (Zhou et al., 2021), making a strong reason for the dominant position of DO in LTSF. However, the shortcomings of the DO approach itself have been largely overlooked. We argue that despite its simplicity, the most significant weakness of DO, particularly when compared to AR, is its lack of inherent causality.

To illustrate this, we conduct a simple experiment in Fig.3: we consecutively split a forecast of length $L$ into $m$ non-overlapped sub-forecasts with length $L/m$, and each part is predicted by an independent linear head from a shared representation. The final forecast is a concatenation of these segments. Empirically, it turns out that the performance of this multi-headed model is nearly identical to a standard DO model with a single head predicting the full horizon. This result suggests that the NF is unaware of the temporal relationships within the future sequence and makes the prediction without considering the distance/interval between the input history and output future. As a result, it learns to forecast each segment independently, which is fundamentally anti-intuitive.

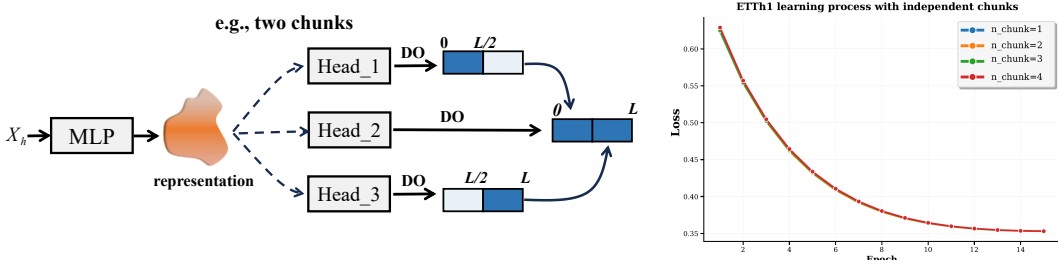

Figure 3: We make the forecast by applying independent heads on several non-overlapped chunks. The right figure shows the learning process in different settings.

According to this observation, we propose a new forecasting paradigm as follows,

**Definition 1** (Boost Direct Output (BDO)). *Given a history time series $T_x = \{x_1, x_2, \cdots, x_h\}$ (abbr. $T_x\{1 : h\}$) in length $h$, the NF recursively generates an estimation $\hat{T}_y$ of the future object $T_y = \{y_1, y_2, \cdots, y_L\}$ over $N$-steps. Let $\hat{T}_y\{1 : h_{n-1}\}$ be the forecast at step $n-1$, then the forecast at step $n$ is:*

$$\hat{T}_y\{1 : h_n\} = NF([T_x, \hat{T}_y\{1 : h_{n-1}\}]), \ n = 1, 2, \cdots, N; \tag{2}$$

where $h_0 = 1$ and $h_N = y_L$. $[\cdot, \cdot]$ indicates the concatenation along the temporal dimension, $L$ is the entire prediction length. In our implementation, we evenly split the forecasting length into $n$ segments for convenience, i.e., in Eq. 2, $h_N = h * N$, where $N$ is a factor of $L$.

Intuitively, BDO recursively generates forecasts for progressively longer horizons, reusing predictions from previous stages. This incorporates an AR-like causal structure into the forecasting process, while retaining the patch-wise output characteristic of DO that mitigates significant error accumulation. A balance between these two properties can be achieved by properly setting the number of recursive stages, $N$. Furthermore, since short-term forecasting is generally an easier task than long-term forecasting, BDO effectively creates a learning curriculum and tends to build hierarchical representations as illustrated in Fig.2. We can reasonably expect this paradigm to outperform both pure AR and pure DO strategies, especially in the challenging LTSF setting.

## 2.4 MODEL ARCHITECTURE

We construct our NFs using only MLP and linear layers for three primary reasons: 1) As foundational deep learning modules, improvements demonstrated on them are broadly applicable and convincing. 2) Their computational efficiency (low FLOPs) facilitates rapid experimentation and verification. 3) Recent work has shown that simple MLP-based architectures are often sufficient for a wide range of forecasting tasks (Ekambaram et al., 2023).

In principle, we build two NF variants, ReNF-$\alpha$ and ReNF-$\beta$, which differ in their degree of non-linearity to better handle datasets of varying complexity. The overall architecture is shown in Fig.4. We stack multiple blocks to adopt the BDO strategy. Each consists of a linear layer for representation projection, followed by an MLP for nonlinear transformation. Each block is also equipped with

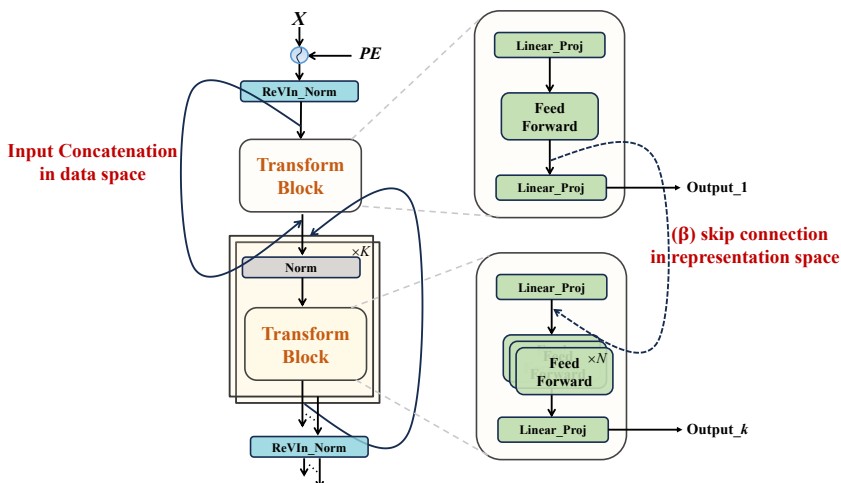

Figure 4: Model Structure of ReNF.

a dedicated linear head that maps the internal representation back to the data space, steering it to function as a sub-forecaster for horizon $h_j, j = 0, 1, ..., k$. In the recursive BDO process, the output of each sub-forecaster is concatenated with the original input data to form the new input for the next one. To allow for deeper representation flow, the ReNF-$\beta$ variant also incorporates skip-connections between the representation spaces of consecutive blocks.

We employ RevIN (Kim et al., 2021) as the pre-normalization for the initial input data to reduce the distribution discrepancy between the training and evaluation phases. For consistency, we also apply pre-normalizations to the input of each sub-forecaster. Additionally, we apply dropout before the initial linear projection to prevent the model from being overly dependent on the history observation and the concatenated information in the data space.

The general equation for the transformation process within each sub-forecaster can be written as follows,

$$\text{ReNF\_Block}(T) = \text{Proj}(\text{Transform}(\text{Proj}(\text{Norm}(\text{Drop}(T))))) \tag{3}$$

where $T$ denotes the input series, and all the projections are performed on the temporal dimension.

## 2.5 Learning Objective

To train a deterministic NF, we adopt the hybrid loss function as used in (Liu et al., 2025a), which is a convex combination of the Mean Absolute Error (MAE) in both the time and frequency domains. The frequency-domain component has been shown to be effective at reducing spurious autocorrelations in the labels (Wang et al., 2024a).

Our BDO framework uniquely generates multiple outputs of varying lengths, enabling us to apply this loss at each forecasting stage. This hierarchical supervision allows us to fully leverage the label information at multiple scales, encouraging the model to build causally structured and homogeneous representations rather than the disconnected ones typical of standard DO forecasts. In the words of our MNFT, this multi-level supervision provides richer feedback and mutual information from the observed labels, thereby constraining the solution space more effectively.

In the BDO paradigm, the quality of early, short-term forecasts is critical, as errors at these stages may still propagate and degrade the performance of subsequent long-term forecasts. To ensure a stable foundation, we therefore place heavier weights on the losses computed at earlier stages (i.e., for shorter forecast horizons). The complete loss function is expressed as follows:

$$loss = \sum_{n=1}^{N} (\gamma/n) * (\alpha * \|\hat{Y}_f^{(n)} - X_f^{(n)}\|_1 + (1 - \alpha) * \|\text{Freq}(\hat{Y}_f^{(n)}) - \text{Freq}(X_f^{(n)})\|_1) \tag{4}$$

where $\| \cdot \|_1$ denotes the $l_1$ norm, $\gamma$ and $\alpha$ is predifined coefficients. Freq$(\cdot)$ represents the discrete fourier transform and $\hat{Y}_f^{(n)}$ denotes the $n$-th sub-forecast corresponding to the $n$-th piece of $X_f$. A more in-depth analysis of this loss function is provided in Appendix D.

## 2.6 Smoothing the Learning for Time Series

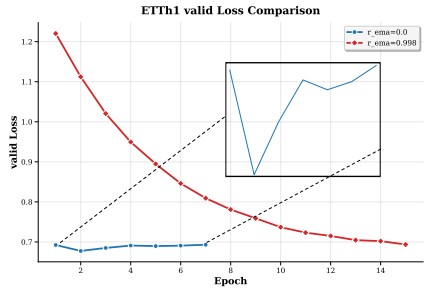 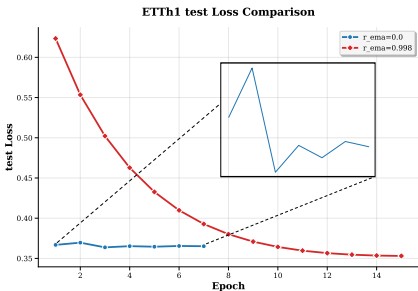

Figure 5: Variation of valid and test loss before and after applying EMA smoothing. The valid loss and test loss are not consistent during the learning process of NFs without smoothing.

Notably, training on standard benchmark datasets is often unstable, partly due to sparse or redundant information. The non-stationary nature of real-world time series, combined with chronological data splitting, leads to both internal (batch-to-batch) and external (train/validation/test) distribution mismatches. This causes the optimization path to be heterogeneous, rendering the learning process for NFs ineffective. Specifically, internal data redundancy can cause the model to overfit repeated or spurious patterns (Liu et al., 2025a), harming generalization, while external distribution shift creates inconsistencies across the different phases of training. While normalization techniques like RevIN address the train-test shift, a critical problem remains between validation and testing.

On many datasets, particularly those with smaller volumes, the validation and test losses exhibit significantly different, and at times conflicting, dynamics during training. This issue invalidates the early stopping criterion: learning steps become ineffective, and suboptimal models are saved, preventing a true assessment of a model's capabilities. This is especially problematic when comparing models of varying complexities, which naturally have different optimization trajectories.

To mitigate these unexpected effects, we propose smoothing the training trajectory by employing a "shadow" model for evaluation. We achieve this efficiently using an Exponential Moving Average (EMA) to track the parameters of the online model, a technique proven effective in self-supervised learning in self-supervised learning (LeCun, 2022) and generative models (Song et al., 2020). Specifically, let $\theta$ be the parameters of the online model being trained, and $\theta'$ be the parameters of the shadow model. Then the shadow model's parameters are upgraded as follows at each iteration.

$$\theta'_{\text{new}} = \alpha * \theta'_{\text{prev}} + (1 - \alpha) * \theta_{\text{current}} \tag{5}$$

where $\alpha$ is the EMA decay rate. This shadow model is then used for all evaluations. As shown in Fig. 5, this technique effectively smooths the learning curves and mitigates the inconsistency between validation and test performance. By providing a more stable and reliable training signal, EMA prolongs the effective learning period and enables the model to converge to better local minima, achieving improved generalization.

## 3 Experiment

**Baselines and Datasets.** We compare our model with a suite of recent state-of-the-art methods, including TimeBridge (Liu et al., 2025a), DUET (Qiu et al., 2025), TimeDistill (Ni et al., 2025), Timer-XL (Liu et al., 2025b), iTransformer (Liu et al., 2024), TimeMixer (Wang et al., 2024b), PatchTST (Nie et al., 2023), Crossformer (Zhang & Yan, 2023), and Dlinear (Zeng et al., 2023). We use widely used benchmark datasets in this area, including electricity (ETTh1, ETTh2, ETTm1, ETTm2, Electricity), environment (Weather), energy (Solar-Energy), and transportation (Traffic). Supplementary datasets for the evaluation on short-term forecasting and the detailed descriptions of these datasets are included in Appendix B.

**Setups.** All experiments were conducted on a single NVIDIA 4090 GPU with 24GB of memory, using the Adam optimizer (Kingma & Ba, 2014) and a fixed random seed of 2021 for reproducibility. Results for all baseline models were reproduced using their official source code and optimal configurations.

For our model, ReNF, we searched for the optimal learning rate in the range from 0.0001 to 0.005, the EMA decay rates in the range from 0.99 to 0.999, and the number of layers from 2 to 8. We apply ReNF-$\alpha$ to ETTh1 and ETTh2 datasets, and ReNF-$\beta$ to others. The look-back window of ReNF is searched over $\{336, 512, 720\}$ for the best performance. The drop_last bug is corrected following the TFB benchmark (Qiu et al., 2025).

## 3.1 MAIN RESULT

The results, presented in Table 2, demonstrate a powerful conclusion: without resorting to complex, specialized modules for multi-resolution, periodicity, or cross-variate dependencies, a simple MLP-based model can achieve exceptional performance. **By focusing instead on fundamentally improving the forecasting paradigm and stabilizing the training process**, our MLP model-ReNF sets a new state-of-the-art. Overall, ReNF surpasses all leading 2024 models by a significant margin and outperforms even the very competitive 2025 SOTA methods in almost all cases. This provides strong evidence for the effectiveness of our proposed techniques.

However, we do not claim that existing specialized techniques are redundant. Rather, as discussed in Sec.2.6, many of these architectural designs were evaluated within unstable training frameworks, suggesting that their true effects may need tedious re-evaluation, which may be a promising future work. For instance, on the complex Traffic dataset, while ReNF shows significant improvement over many baselines, it does not surpass the MSE score of the large Transformer-based model, TimeBridge. We compare the model complexity in FLOPs and params between the most competitive models in the Table 1, showing that ReNF reduces 20x the complexity in FLOPs compared to TimeBridge with the Traffic dataset. This suggests that more complex models might still work better on datasets with high non-linearity.

| Model | | ReNF | TimeBridge | DUET |
|---|---|---|---|---|
| Weather | Params (MB) | **0.200** | 0.887 | 4.128 |
| | FLOPs (GB) | **0.004** | 4.262 | 0.143 |
| ETTm2 | Params (MB) | **0.393** | 0.460 | 4.658 |
| | FLOPs (GB) | **0.003** | 1.604 | 0.051 |
| Traffic | Param (MB) | 21.476 | 12.431 | **9.910** |
| | FLOPs (GB) | 22.813 | 479.231 | **15.575** |

Table 1: Efficiency comparison of ReNF, TimeBridge, and DUET. All metrics are averaged across the four prediction lengths.

| Models | ReNF ours | | TimeBridge ((2025a)) | | DUET (2025) | | TimeDistill (2025) | | Timer-XL (2025b) | | iTransformer (2024) | | TimeMixer (2024b) | | PatchTST (2023) | | Crossformer (2023) | | DLinear (2023) | |
|---|---|---|---|---|---|---|---|---|---|---|---|---|---|---|---|---|---|---|---|---|
| Metric | MSE | MAE | MSE | MAE | MSE | MAE | MSE | MAE | MSE | MAE | MSE | MAE | MSE | MAE | MSE | MAE | MSE | MAE | MSE | MAE |
| Weather | **0.214** | **0.247** | 0.220 | 0.250 | 0.219 | 0.253 | 0.221 | 0.269 | 0.240 | 0.273 | 0.232 | 0.269 | 0.226 | 0.264 | 0.224 | 0.262 | 0.232 | 0.294 | 0.242 | 0.293 |
| Electricity | **0.145** | **0.237** | 0.152 | 0.247 | 0.157 | 0.248 | 0.157 | 0.254 | 0.155 | 0.246 | 0.163 | 0.258 | 0.185 | 0.284 | 0.171 | 0.271 | 0.171 | 0.263 | 0.167 | 0.264 |
| Traffic | 0.365 | **0.245** | **0.357** | 0.248 | 0.393 | 0.256 | 0.387 | 0.271 | 0.374 | 0.255 | 0.395 | 0.279 | 0.409 | 0.279 | 0.397 | 0.275 | 0.522 | 0.282 | 0.418 | 0.287 |
| Solar | **0.176** | **0.214** | 0.183 | 0.219 | 0.195 | 0.214 | 0.184 | 0.242 | 0.198 | 0.249 | 0.202 | 0.262 | 0.193 | 0.252 | 0.200 | 0.284 | 0.205 | 0.233 | 0.224 | 0.286 |
| ETTm1 | **0.331** | **0.364** | 0.349 | 0.380 | 0.338 | 0.369 | 0.348 | 0.380 | 0.359 | 0.382 | 0.361 | 0.390 | 0.356 | 0.380 | 0.349 | 0.381 | 0.464 | 0.456 | 0.356 | 0.379 |
| ETTm2 | **0.243** | **0.301** | 0.247 | 0.305 | 0.248 | 0.308 | 0.250 | 0.312 | 0.271 | 0.322 | 0.269 | 0.327 | 0.257 | 0.318 | 0.256 | 0.314 | 0.501 | 0.505 | 0.259 | 0.325 |
| ETTh1 | **0.391** | **0.416** | 0.401 | 0.426 | 0.401 | 0.420 | 0.430 | 0.441 | 0.409 | 0.430 | 0.439 | 0.448 | 0.427 | 0.441 | 0.419 | 0.436 | 0.439 | 0.461 | 0.425 | 0.439 |
| ETTh2 | **0.327** | **0.379** | 0.345 | 0.386 | 0.336 | 0.385 | 0.345 | 0.395 | 0.352 | 0.402 | 0.370 | 0.403 | 0.349 | 0.397 | 0.351 | 0.395 | 0.894 | 0.680 | 0.470 | 0.468 |

Table 2: Results of long-term forecasting of hyperparameter searching. All results are averaged across four different prediction lengths: $\{96, 192, 336, 720\}$. The best and second-best results are highlighted in red and blue, respectively. Full results are listed in Appendix E.

## 3.2 ABLATION STUDY

**Effect of EMA.** To analyze the impact of EMA, we recorded the evaluation dynamics of ReNF with and without our smoothing technique. The results, shown in Fig. 6, clearly demonstrate EMA's role. It is notable that we also record the variation of test loss as evidence for the effect of improving the generalization ability.

First, on smaller or less stable datasets like ETTh2, EMA mitigates spurious overfitting and stabilizes the learning curves. This provides a more reliable signal for early stopping and prolongs the effective training period. Second, on large, high-quality datasets such as Electricity, the choice of EMA decay rate can influence performance. By selecting an appropriate rate, the generalization ability of the NF can be substantially enhanced. Full numerical results for all datasets are available in the Appendix.

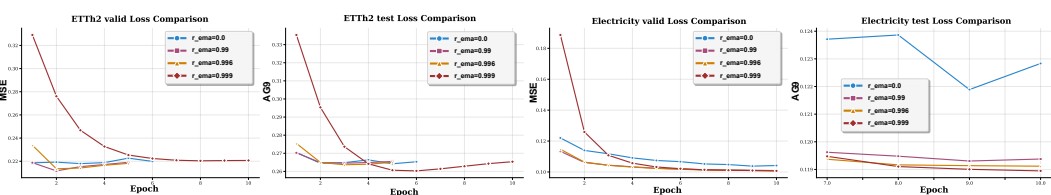

Figure 6: Effect of the EMA smoothing on training and test phase.

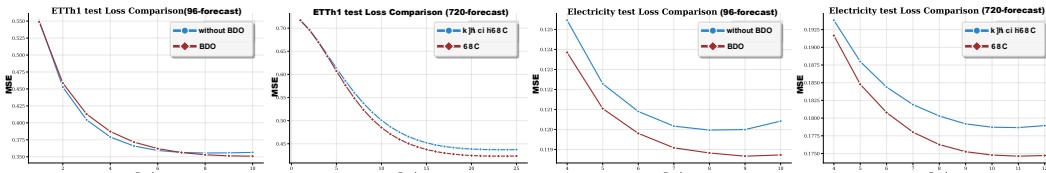

Figure 7: Effect of the BDO forecast.

**Effect of BDO.** We investigate the effect of the BDO paradigm with the following settings: 1) Keep the depth of ReNF, but disable the recursive input concatenation and apply the loss function only to the final output; 2) Directly change the number of layers/sub-forecasts.

By config.1, the BDO reduces to DO with the model structure unchanged. The results, shown in Fig. 7, demonstrate that even on the ETTh1 dataset, which has limited data and is difficult to optimize with deep representations, our BDO strategy can still yield superior performance, especially for very long-term forecasts. This finding empirically supports our claim, derived from proposition 2.2, that leveraging multiple, hierarchically-generated sub-forecasts provides valuable mutual information that enhances the final prediction.

| Layer | | K=1 | K=2 | K=3 | K=4 | K=5 | K=6 |
|-------|-----|-----|-----|-----|-----|-----|-----|
| Weather | MSE | 0.311 | 0.309 | 0.307 | 0.307 | 0.307 | 0.307 |
| | MAE | 0.322 | 0.320 | 0.319 | 0.319 | 0.319 | 0.319 |
| ETTm1 | MSE | 0.411 | 0.408 | 0.407 | 0.406 | 0.404 | 0.401 |
| | MAE | 0.411 | 0.409 | 0.408 | 0.407 | 0.407 | 0.407 |
| Traffic | MSE | 0.426 | 0.415 | 0.410 | 0.406 | 0.403 | 0.402 |
| | MAE | 0.281 | 0.274 | 0.272 | 0.270 | 0.267 | 0.267 |

Table 3: 720-length forecast with varying number of layers (sub-forecasts).

Furthermore, it is shown clearly in Fig. 8(a) that the performance of ReNF with the BDO strategy can consistently improve as K increases. In stark contrast, the performance of the DO model stagnates or degrades with added depth. This difference highlights BDO's ability to effectively utilize deeper architectures. This meaningful phenomenon suggests that our paradigm may unlock a new, more effective scaling law for LTSF models. Diverse examples are shown in the Table 3 and the Appendix E.

## 3.3 EMPIRICAL EVALUATION OF THE PROPOSITION

| | Dataset | ETTm1 | ETTm2 | ETTh1 | ETTh2 | Weather | Electricity | Solar | Traffic |
|------|---------|-------|-------|-------|-------|---------|-------------|-------|---------|
| ReNF | Last Forecast | 0.331 | 0.243 | 0.391 | 0.327 | 0.214 | 0.145 | 0.176 | 0.365 |
| | Empirical Bound | 0.225 | 0.201 | 0.270 | 0.252 | 0.165 | 0.100 | 0.090 | 0.254 |

Table 4: Comparison between the last single forecast and the optimal post-combined forecast of ReNF. The shown metrics are MSE and are averaged across the four prediction lengths.

To empirically evaluate our Multiple Neural Forecasting Proposition, we implement the post-combination strategy described in Sec. 2.2. This requires defining a combination function, $g_c$. While finding an optimal solution without future knowledge is difficult, we can establish a theoretical upper bound on performance. Given the ground truth labels, a trivial yet optimal strategy is to select, at each timestep, the forecast value from among all candidates that has the smallest error. By applying this "oracle" combination, we aim to explore two points: 1) to quantify the potential accuracy improvement achievable through post-combination, thereby establishing a dynamic empirical bound for our forecasting model; and 2) to verify that this empirical bound behaves in a manner consistent with the intuitions of the MNFP 2.2.

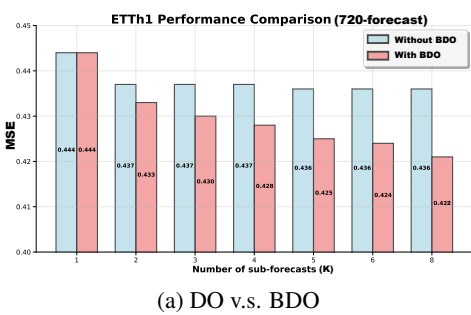 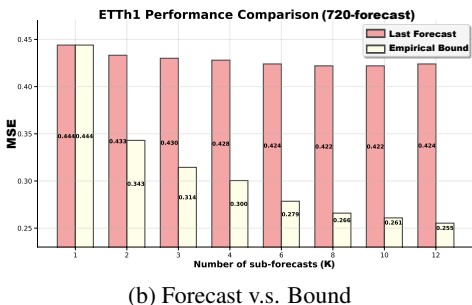

(a) DO v.s. BDO            (b) Forecast v.s. Bound

Figure 8: Performance variation with different numbers of sub-forecasts K. (a). Comparison of using DO or BDO. (b).Comparison between the best forecast and the empirical bound of ReNF.

Table 4 presents the performance of this oracle post-combination strategy, revealing a significant gap between the final ReNF forecast and the empirically optimal combination. While this oracle performance is unattainable in practice, the resulting empirical bound is highly informative. On one hand, it indicates that any single forecast is suboptimal and that powerful combinatorial strategies for improving predictions must exist (a promising direction for future work). On the other hand, it shows that we are still far from perfectly leveraging the information contained within the multiple sub-forecasts. This gap is precisely what motivates our BDO paradigm, which is designed to help the neural network implicitly learn a more effective combination function $g_c$.

Furthermore, Fig. 8(b) shows how this empirical bound varies with the number of sub-forecasts $K$. The bound consistently decays as $K$ increases, even when the model's performance saturates. This result aligns perfectly with the core insight from MNFP: the theoretical performance limit improves as the number of candidate forecasts grows.

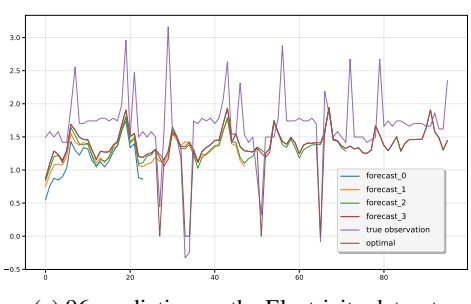 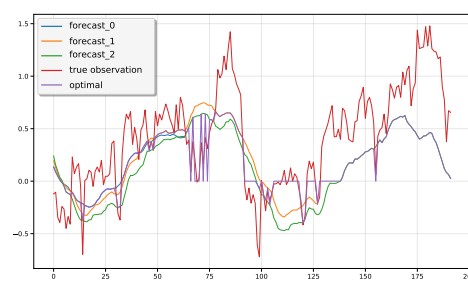

(a) 96-prediction on the Electricity dataset    (b) 192-prediction on the ETTm2 dataset

Figure 9: Visualization of forecasting results of ReNF. The figure shows multiple outputs of ReNF in different layers, along with the result of applying optimal post-combination.

In a nutshell, our evaluation of the MNFP clarifies two primary roles for Neural Networks in this domain. First, NNs should be leveraged to learn a powerful post-combination function over multiple candidate forecasts. Second, more powerful base forecasters must be developed to better approximate the true data-generating distribution, directly reducing the predictive bias $b$ as identified in our proposition 2.2.

## 4 CONCLUSION

We propose a fundamental proposition to support the mechanism of multiple forecasts. Based on this proposition, we introduce a novel paradigm that combines the advantages of current forecasting methods, enabling more accurate predictions in a hierarchical representation space. To remedy training instability, we leverage EMA to smooth the learning process of neural forecasters. Our empirical results show that with these techniques, a simple MLP can outperform recent SOTA models with significantly less complexity. The value of our proposition is also evaluated and confirmed by experimental results, clearly identifying the role of the Neural Network in building the Long-Term Time Series forecasters.

## 5 REPRODUCIBILITY STATEMENT

We have put a lot of effort into ensuring the reproducibility of this work, including the code in the anonymous repository `https://anonymous.4open.science/r/ReNF-A151`, a complete data description in the Appendix B, and clear explanations of the assumptions and proof of the proposition in the Appendix C.

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

# Supplementary Material

## A    RELATED WORK

We note that ReNF is not the first work to address LTSF using multiple short-term forecasts. A pioneering and highly relevant study by (Nguyen & Chan, 2004) proposed a machine called MNN, which generates long-term forecasts using multiple neural networks, each responsible for a specific interval of the output. The essential distinction between MNN and our approach is that we generate sub-forecasts recursively, with the explicit goal of injecting causality into the DO strategy. In contrast, each sub-network in MNN functions as a standalone AR(1) model. This difference stems from our distinct motivation: whereas MNN was designed primarily to mitigate the error accumulation of one-step-ahead NN models, our work begins with a foundational forecasting proposition. Based on this proposition, we leverage modern deep learning techniques to effectively utilize multiple sub-forecasts, achieving strong empirical results on a wide range of real-world datasets, which constitutes one of our main contributions.

A recently proposed work using the multiple-choice learning paradigm (Cortés et al., 2025) is also relevant, as it demonstrates an ability to handle the diverse and multi-modal nature of the future from a probabilistic perspective. Furthermore, the characterization of TimeMCL as a conditional stationary quantizer for time series may offer additional theoretical support and interpretations for our framework.

**Ensemble.** Our work also connects to classic ensemble methods in machine learning (Mohri et al., 2018). For instance, the recursive workflow of ReNF is analogous to gradient-boosted regression models (Chen & Guestrin, 2016), which construct a strong predictor from multiple weak ones. Additionally, the collaborative training can be viewed as a form of bagging (Breiman, 1996), which effectively resamples the data to train diverse forecasters. In this context, an even more recent study (Modi & Pan, 2025) re-certified the benefits of such ensemble methods for enhancing Transformer-based NFs, providing further empirical support for the direction of our research.

**Forecast Combinations.** Forecast combination is a classic technique for improving forecast accuracy and robustness by leveraging the diverse strengths of multiple models (Clemen, 1989). In this work, we extend this concept (Bates & Granger, 1969) to the domain of deep learning for long-term time series forecasting. Rather than combining distinct, parallel forecasters into a hybrid model (Zhang, 2003), our framework achieves this goal efficiently within a single, structured approach for generating and implicitly combining forecasts within a single neural network. By recursively stacking sub-forecasts, similar in spirit to N-BEATS (Oreshkin et al., 2019) and NHits (Challu et al., 2023) which ensure the frequency diversity of the forecasts in different stages, we learn and compose multiple forecasts in different lengths while the representation is both deep and diverse, allowing the LSTF task to benefit from the full power of modern machine learning.

## B    DETAILS OF DATASETS

The ETTh1, ETTh2, ETTm1, and ETTm2 datasets record the temperature of electricity transformers every hour and every 15 minutes. The Weather dataset contains 21 weather information, measured every 10 minutes in Germany. The Electricity dataset records the amount of electricity used by 321 customers every hour. The Solar dataset records how much electricity is produced by solar power stations every 10 minutes, from 137 solar power stations, in 2006. The Traffic dataset records how busy the roads are in San Francisco, every hour, from 862 sensors on the freeway.

In addition, we also evaluate the short-term time series forecasting on six datasets that are used in the study (Yue et al., 2025). The METR-LA database was populated with traffic network data in Los Angeles during the springtime period of 2012, specifically from March to June. This data was collected at an interval of five minutes. The NASDAQ includes the daily NASDAQ index and key economic indicators from 2010 to 2024. The SP500 records daily SP500 index data (e.g., opening price, closing price, and trading volume) from January 1993 to February 2025. CarSales collects

| Dataset | Dim | Prediction Length | Split | Frequency | domain |
|---------|-----|-------------------|-------|-----------|--------|
| ETTh1, ETTh2 | 7 | 96, 192, 336, 720 | (6, 2, 2) | Hourly | Electricity |
| ETTm1, ETTm2 | 7 | 96, 192, 336, 720 | (6, 2, 2) | 15min | Electricity |
| Weather | 21 | 96, 192, 336, 720 | (7, 1, 2) | 10min | Environment |
| Electricity | 321 | 96, 192, 336, 720 | (7, 1, 2) | Hourly | Electricity |
| Traffic | 862 | 96, 192, 336, 720 | (7, 1, 2) | Hourly | Transportation |
| Solar | 137 | 96, 192, 336, 720 | (6, 2, 2) | 10min | Energy |
| NASDAQ | 12 | 24, 36, 48, 60 | (7, 1, 2) | Daily | Finance |
| SP500 | 12 | 24, 36, 48, 60 | (7, 1, 2) | Daily | Finance |
| Carsales | 12 | 24, 36, 48, 60 | (7, 1, 2) | Daily | Market |
| Website | 12 | 24, 36, 48, 60 | (7, 1, 2) | Daily | Web |
| Power | 12 | 24, 36, 48, 60 | (7, 1, 2) | Daily | Energy |
| METR-LA | 207 | 96, 192, 336, 720 | (7, 1, 2) | 5min | Transportation |

Table 5: Descriptions of multivariate time series datasets used in this research. The dim column represents the number of variates, and the split column specifies the train-validate-test splitting ratio for each dataset.

daily sales of 10 vehicle brands (e.g., Toyota, Honda) in the U.S. from January 2005 to June 2023. The data are compiled from the Vehicle Sales dataset on Kaggle. Power contains daily wind and solar energy production (in MW) records for the French grid from April 2020 to June 2023. The data are compiled from the Wind and Solar Daily Power Production dataset on Kaggle. The website contains six years of daily visit data (e.g,. first-time and returning visits) to an academic website, spanning from September 2014 to August 2020.

## C   PROOF OF THE APPROXIMATE PROPOSITION

We restate the parameters and preconditions to derive the target bound value:

- (A1) The objective time series with finite length is bounded by a positive real constant $\lambda$.

- (A2) The NFM generates candidate series $\{\hat{Y}_f^{(i)}\}_{i=1}^{c>1}$ under various $\theta^i$ or random states $\gamma^i$.

- (A3) The predictive bias of the NFM is bounded by up to $\sup_t |\hat{\mu}_t - \mu_t| = b$.

*Proof.* First, we derive the expected bound for each independent element $y_t^{(i)}$ in any candidate forecast $\hat{Y}_f^{(i)} = \{y_1^{(i)}, y^{(i)}, \cdots, y_c^{(i)}\}$.

Since $y_t^{(i)} \sim \hat{p}(\hat{\mu}_t, \hat{\sigma}_t^2)$, we know from the (A2) that $\{y_t^{(i)}\}_{i=1}^c$ are i.i.d. samples from the same distribution with $\mathbb{E}(y_t^{(i)}) = \hat{\mu}_t$. Thus, according to the law of large numbers, the statistical average of $\{y_t^{(i)}\}_{i=1}^c$ converges to the expectation $\mu_t$ almost surely as $c \to \infty$.

In particular, we have

$$\mathbb{E}|\hat{\mu}_t - \frac{1}{c}\sum_{i=1}^c y_t^{(i)}|^2 = \mathbb{E}|\frac{1}{c}\sum_{i=1}^c (y_t^{(i)} - \hat{\mu}_t) + \hat{\mu}_t - \hat{\mu}_t|^2 \tag{6}$$

$$= \frac{1}{c^2}\mathbb{E}|\sum_{i=1}^c (y_t^{(i)} - \hat{\mu}_t)|^2 \tag{7}$$

$$= \frac{1}{c^2}\sum_{i=1}^c \mathbb{E}|y_t^{(i)} - \hat{\mu}_t|^2. \tag{8}$$

The last identity holds because $\{y_t^{(i)}\}_{i=1}^c$ are independent, i.e., $\mathbb{E}[(y_t^{(i)} - \hat{\mu}_t)(y_t^{(j)} - \hat{\mu}_t)] = \mathbb{E}(y_t^{(i)} - \hat{\mu}_t) \cdot \mathbb{E}(y_t^{(j)} - \hat{\mu}_t) = 0$, $i \neq j$.

Then, we can bound this term by showing

$$\mathbb{E}|y_t^{(i)} - \hat{\mu}_t|^2 = \mathbb{E}|y_t^{(i)} - \mathbb{E}[y_t^{(i)}]|^2 \tag{9}$$

$$= \mathbb{E}|(y_t^{(i)})^2| - |\mathbb{E}[y_t^{(i)}]|^2 \text{ (variance identity)} \tag{10}$$

$$\leq \mathbb{E}[(y_t^{(i)})^2] \tag{11}$$

$$\leq \sup_{i,t}\{(y_t^{(i)})^2\} = \lambda^2. \text{ (A1)} \tag{12}$$

Therefore, we have shown that

$$\mathbb{E}|\hat{\mu}_t - \frac{1}{c}\sum_{i=1}^{c} y_t^{(i)}|^2 \leq \frac{\lambda^2}{c}. \tag{13}$$

for any $i$; and in particular, there must exist a set of $\{\tilde{y}_t^{(i)}\}_{i=1}^{c}$ satisfying

$$|\hat{\mu}_t - \frac{1}{c}\sum_{i=1}^{c} \tilde{y}_t^{(i)}| \leq \frac{\lambda}{\sqrt{c}}. \tag{14}$$

Fixing the set $\{\tilde{y}_t^{(i)}\}_{i=1}^{c}$, it remains to estimate the distance between the forecast by the NFM $\Phi$ and the expected forecast $Y_f$, and the true observation $X_f$, respectively.

Since it is trivial to get

$$|\mu_t - \frac{1}{c}\sum_{i=1}^{c} \tilde{y}_t^{(i)}| \leq |\hat{\mu}_t - \frac{1}{c}\sum_{i=1}^{c} \tilde{y}_t^{(i)}| + |\hat{\mu}_t - \mu_t| \leq |\hat{\mu}_t - \frac{1}{c}\sum_{i=1}^{c} \tilde{y}_t^{(i)}| + b. \text{ (A3)} \tag{15}$$

We have

$$|x_t - \frac{1}{c}\sum_{i=1}^{c} \tilde{y}_t^{(i)}| \leq |\mu_t - \frac{1}{c}\sum_{i=1}^{c} \tilde{y}_t^{(i)}| + |x_t - \mu_t| \leq |\hat{\mu}_t - \frac{1}{c}\sum_{i=1}^{c} \tilde{y}_t^{(i)}| + b + \sigma. \tag{16}$$

Thus, we can derive the expected upper bound as

$$\sum_{t=1}^{T}|x_t - \frac{1}{c}\sum_{i=1}^{c} \tilde{y}_t^{(i)}| \leq T \cdot (\frac{\lambda}{c} + b + \sigma) = \frac{T(\lambda + \sqrt{c}(b + \sigma))}{\sqrt{c}}. \tag{17}$$

This completes the proof.

### C.1 VARIANCE ANALYSIS OF TOTAL SUMMED ERROR

The above proof gives an intuitive bound, which does not depend on any assumption on the temporal dependence. In this part, we consider this factor as a pivot for analysing the effects of different forecasting paradigms, identifying the role of our BDO.

The variance of this total error is:

$$var(\sum_{t=1}^{T} e_t) = \sum_{t=1}^{T} var(e_t) + 2\sum_{t<t'} Cov(e_t, e_t') \tag{18}$$

For simplicity, let the prediction error at each timestep as $t$ $e_t := \mu_t - \frac{1}{c}\sum_{i=1}^{c} \tilde{y}_t^{(i)}$, and temporarily omit the effect of the predictive bias $b$ of NFM in A3, i.e., $\mu = \hat{\mu}$. Then we can compute the covariance (autocorrelation) of the errors at any two distinct timesteps $k$ and $h$,

$$Cov(e_k, e_h) = \mathbb{E}[(\mu_k - \tilde{\mu}_k)(\mu_h - \tilde{\mu}_h)] \tag{19}$$

$$= \mathbb{E}[\tilde{\mu}_k\tilde{\mu}_h] - \mu_k\mu_h \tag{20}$$

$$= Cov(\tilde{\mu}_k\tilde{\mu}_h) \tag{21}$$

where we denote $\tilde{\mu}_t := \frac{1}{c} \sum_{i=1}^{c} \tilde{y}_t^{(i)}$. So we have

$$Cov(\tilde{\mu}_k \tilde{\mu}_h) = \frac{1}{c^2} \sum_{i=1}^{c} \sum_{j=1}^{c} Cov(y_k^{(i)}, y_h^{(j)}) \tag{22}$$

$$= \frac{1}{c} Cov(y_k^{(i)}, y_h^{(i)}), \ \forall i \tag{23}$$

The last inequality holds by assumption A2, which isolates the correlations across different candidates.

Now we can consider the role of temporal dependence in the error structure.

**Case 1: Temporal Independence (The DO Paradigm):** The Direct Output (DO) paradigm is designed to approximately satisfy the assumption of temporal independence, as evidenced in Sec. 2. This implies that the error covariance terms in Eq. 23 are zero. Consequently, the total error variance is simply the sum of the per-timestep variances, While this structure effectively prevents the compounding of errors, it often comes at the cost of the model being "unaware" of the sequential dynamics within the future horizon, potentially leading to higher per-timestep variance. Furthermore, the variance of the total summed error can only benefit from the combination factor $c$ in reducing the individual uncertainty at each future timestep $\sum_{t=1}^{T} var(e_t)$.

**Case 2: Temporal Dependence (The AR Paradigm).** : The typical AR paradigm fundamentally violates this assumption and thus leads to errors from one step propagating to the next, which causes the total variance to explode over long horizons. Even though the introduction of the ensembling factor c would reduce the magnitude of this effect, it does not alter the underlying structural problem of error accumulation.

**Case 3: A Synthesis (The BDO Paradigm).**: Our Boosted Direct Output paradigm operates as a synthesis of these two extremes. It intentionally violates strict temporal independence by recursively feeding sub-forecasts back into the model, thereby making it aware of causal dependencies within the forecast horizon. However, unlike a pure AR model, it would learn a mapping that minimizes the error covariance. Through hierarchical supervision and its patch-wise output structure inherited from DO, the model is trained to control the accumulation of errors. The goal is to leverage the benefits of modeling temporal structure while constraining the error correlation, effectively learning to make the covariance term in Eq. 23 as small as possible.

Note that the above proof and analysis are based on a univariate time series; however, it is easy to extend the result to the multivariate case using a similar process in a certain normed vector space.

## D  IMPLICIT STRUCTURAL LOSS

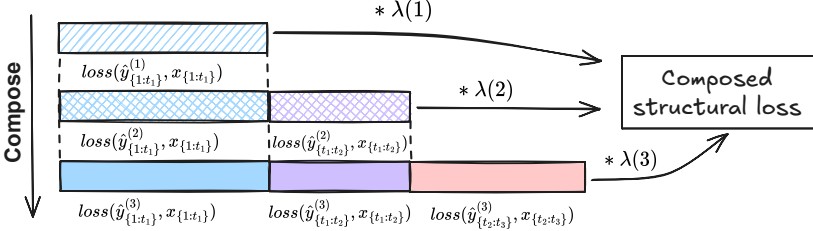

Figure 10: Illustration of the implicit structural loss of BDO, we exemplify it in the NF consisting of three sub-forecasters.

An interesting property of the BDO paradigm is the implicit structural loss it induces. This connects our work to recent research on explicit loss engineering, such as positional weighting (Xue et al., 2023) and patch-wise structural losses (Kudrat et al., 2025). We posit that the BDO learning objective, formed by the weighted sum of losses from hierarchical sub-forecasts, inherently functions as a complex structural loss. This seemingly implicit loss can be seen as the generalized version of the above two.

To formalize this property, we first simplify the loss function from Eq.4 as:

$$Loss = \sum_{n=1}^{N} \lambda(n) * f(\hat{Y}_f^{(n)}, X_f^{(n)}) \tag{24}$$

where $f(\cdot, \cdot)$ denotes a base error function such as the MAE.

Note that in our definitions and notations, $\hat{Y}_f^{(n)} = \{\hat{y}_1^{(n)}, \hat{y}_2^{(n)}, \cdots, \hat{y}_{t_n}^{(n)}\}$. We can therefore expand the total loss into a point-wise sum over all timesteps:

$$Loss = \sum_{n=1}^{N} \sum_{t=1}^{t_n} \lambda(n) * f(\hat{y}_t^{(n)}, x_t) \tag{25}$$

The underlying structure of this composite loss, visualized in Figure 10, reveals a key insight: BDO is not just a recursive forecasting process, but also a method for implicitly constructing a complex and adaptive structural loss. The properties of this loss can be finely tuned through several mechanisms: the stage-wise weighting coefficients, the forecast splitting strategy, the choice of the base error function, and even the architecture of each sub-forecaster.

# E    FULL RESULTS

## E.1    FULL LONG-TERM FORECASTING RESULTS

We present the full version of Table.2 in Table.6 to show the capability of ReNF in Long-Term time series forecasting.

## E.2    SHORT-TERM FORECASTING RESULTS

In addition, to further verify the generality of ReNF, we test it at six supplementary datasets from a recent short-term time series forecasting benchmark (Yue et al., 2025). The descriptions of these datasets can be found in the Table.5. As shown in Table.7, ReNF remains highly competitive against the recent SOTA models in short-term forecasting task. In fact, the short-term forecast can not benefit from the BDO to the extent of LTSF, as we hypothesize that when restricted to shorter look-back windows, the initial short-term forecasts in the BDO process are less accurate. Consequently, concatenating these noisier predictions can introduce disruptions that limit the full benefit of our paradigm. From another perspective, while our proposed methods are universally applicable, these results also highlight the value of more specialized or refined extension of our proposals in further enhancing the capabilities of BDO in various scenarios.

# F    MORE ABLATIONS

## F.1    EMA SMOOTHING

The results in Table 8 confirm that EMA smoothing yields substantial improvements in final forecast accuracy. This empirically validates our hypothesis from Section 2: EMA mitigates the detrimental effects of flawed early stopping, where volatile validation scores lead to the premature saving of suboptimal models. By providing a more reliable and stable training signal, EMA not only enhances performance but also establishes the robust foundation necessary to fairly evaluate our other contributions, such as the BDO paradigm.

## F.2    PRE-DROPOUT.

Our MLP architecture applies dropout to the input data by default as a regularization technique. While its impact can be subtle and dataset-dependent, we perform an ablation study for the sake of completeness. The results of this analysis on the ETTh1, ETTh2, and Electricity datasets are presented in Table 9.

| Models | Re-Bound - | | ReNF ours | | TimeBridge (2025a) | | DUET (2025) | | TimeDistill (2025) | | Timer-XL (2025b) | | iTransformer (2024) | | TimeMixer (2024b) | | PatchTST (2023) | | Crossformer (2023) | | DLinear (2023) | |
|---|---|---|---|---|---|---|---|---|---|---|---|---|---|---|---|---|---|---|---|---|---|---|
| Metric | MSE | MAE | MSE | MAE | MSE | MAE | MSE | MAE | MSE | MAE | MSE | MAE | MSE | MAE | MSE | MAE | MSE | MAE | MSE | MAE | MSE | MAE |
| **Weather** 96 | 0.110 | 0.149 | **0.138** | **0.180** | 0.144 | 0.184 | 0.146 | 0.191 | 0.145 | 0.204 | 0.157 | 0.205 | 0.157 | 0.206 | 0.147 | 0.198 | 0.150 | 0.200 | 0.143 | 0.210 | 0.170 | 0.230 |
| 192 | 0.141 | 0.185 | **0.182** | **0.224** | 0.186 | 0.226 | 0.188 | 0.231 | 0.188 | 0.247 | 0.206 | 0.250 | 0.200 | 0.248 | 0.192 | 0.242 | 0.191 | 0.239 | 0.195 | 0.261 | 0.216 | 0.273 |
| 336 | 0.177 | 0.216 | **0.231** | **0.266** | 0.237 | 0.267 | 0.235 | 0.269 | 0.240 | 0.286 | 0.259 | 0.291 | 0.252 | 0.287 | 0.247 | 0.284 | 0.242 | 0.279 | 0.254 | 0.319 | 258 | 0.307 |
| 720 | 0.232 | 0.261 | **0.304** | **0.318** | 0.312 | 0.321 | 0.308 | 0.319 | 0.310 | 0.338 | 0.337 | 0.344 | 0.320 | 0.336 | 0.318 | 0.330 | 0.312 | 0.330 | 0.335 | 0.385 | 0.323 | 0.362 |
| *Avg.* | 0.165 | 0.203 | **0.214** | **0.247** | 0.220 | 0.250 | 0.219 | 0.253 | 0.221 | 0.269 | 0.240 | 0.273 | 0.232 | 0.269 | 0.226 | 0.264 | 0.224 | 0.262 | 0.232 | 0.294 | 0.242 | 0.293 |
| **Electricity** 96 | 0.083 | 0.171 | **0.118** | **0.210** | 0.122 | 0.216 | 0.128 | 0.219 | 0.128 | 0.225 | 0.127 | 0.219 | 0.134 | 0.230 | 0.153 | 0.256 | 0.143 | 0.247 | 0.134 | 0.231 | 0.140 | 0.237 |
| 192 | 0.097 | 0.186 | **0.138** | **0.229** | 0.143 | 0.238 | 0.145 | 0.235 | 0.145 | 0.241 | 0.145 | 0.236 | 0.154 | 0.250 | 0.168 | 0.269 | 0.158 | 0.261 | 0.146 | 0.243 | 0.154 | 0.251 |
| 336 | 0.105 | 0.196 | **0.151** | **0.244** | 0.163 | 0.258 | 0.163 | 0.255 | 0.161 | 0.258 | 0.159 | 0.252 | 0.169 | 0.265 | 0.189 | 0.291 | 0.168 | 0.267 | 0.165 | 0.264 | 0.169 | 0.268 |
| 720 | 0.115 | 0.207 | **0.173** | **0.266** | 0.178 | 0.274 | 0.193 | 0.281 | 0.195 | 0.291 | 0.187 | 0.277 | 0.194 | 0.288 | 0.228 | 0.320 | 0.214 | 0.307 | 0.237 | 0.314 | 0.204 | 0.301 |
| *Avg.* | 0.100 | 0.190 | **0.145** | **0.237** | 0.152 | 0.247 | 0.157 | 0.248 | 0.157 | 0.254 | 0.155 | 0.246 | 0.163 | 0.258 | 0.185 | 0.284 | 0.171 | 0.271 | 0.171 | 0.263 | 0.167 | 0.264 |
| **Traffic** 96 | 0.234 | 0.167 | 0.335 | **0.226** | 0.332 | 0.237 | 0.360 | 0.238 | 0.358 | 0.256 | 0.340 | 0.238 | 0.358 | 0.258 | 0.369 | 0.257 | 0.370 | 0.262 | 0.526 | 0.288 | 0.395 | 0.275 |
| 192 | 0.245 | 0.175 | 0.356 | 0.239 | **0.343** | **0.239** | 0.383 | 0.249 | 0.374 | 0.264 | 0.360 | 0.248 | 0.382 | 0.271 | 0.399 | 0.272 | 0.386 | 0.269 | 0.503 | 0.263 | 0.407 | 0.280 |
| 336 | 0.254 | 0.180 | 0.366 | **0.246** | **0.360** | 0.249 | 0.395 | 0.259 | 0.389 | 0.271 | 0.377 | 0.256 | 0.396 | 0.277 | 0.407 | 0.272 | 0.396 | 0.275 | 0.505 | 0.276 | 0.417 | 0.286 |
| 720 | 0.284 | 0.197 | 0.402 | **0.267** | **0.392** | 0.268 | 0.435 | 0.278 | 0.428 | 0.292 | 0.418 | 0.279 | 0.445 | 0.308 | 0.461 | 0.316 | 0.435 | 0.295 | 0.552 | 0.301 | 0.454 | 0.308 |
| *Avg.* | 0.254 | 0.180 | 0.365 | **0.245** | **0.357** | 0.248 | 0.393 | 0.256 | 0.387 | 0.271 | 0.374 | 0.255 | 0.395 | 0.279 | 0.409 | 0.279 | 0.397 | 0.275 | 0.522 | 0.282 | 0.418 | 0.287 |
| **Solar** 96 | 0.082 | 0.141 | **0.157** | 0.202 | 0.159 | **0.196** | 0.166 | 0.211 | 0.166 | 0.229 | 0.162 | 0.221 | 0.190 | 0.244 | 0.179 | 0.232 | 0.170 | 0.234 | 0.183 | 0.208 | 0.199 | 0.265 |
| 192 | 0.087 | 0.145 | 0.174 | **0.210** | **0.173** | 0.215 | 0.199 | 0.212 | 0.181 | 0.239 | 0.187 | 0.239 | 0.193 | 0.257 | 0.201 | 0.259 | 0.204 | 0.302 | 0.208 | 0.227 | 0.220 | 0.281 |
| 336 | 0.099 | 0.156 | **0.180** | 0.219 | 0.193 | 0.229 | 0.207 | **0.215** | 0.191 | 0.246 | 0.205 | 0.255 | 0.203 | 0.266 | 0.190 | 0.256 | 0.212 | 0.293 | 0.212 | 0.239 | 0.234 | 0.295 |
| 720 | 0.092 | 0.151 | **0.190** | 0.225 | 0.207 | 0.237 | 0.206 | **0.217** | 0.199 | 0.252 | 0.238 | 0.279 | 0.223 | 0.281 | 0.203 | 0.261 | 0.215 | 0.307 | 0.215 | 0.256 | 0.243 | 0.301 |
| *Avg.* | 0.090 | 0.148 | **0.176** | **0.214** | 0.183 | 0.219 | 0.195 | 0.214 | 0.184 | 0.242 | 0.198 | 0.249 | 0.202 | 0.262 | 0.193 | 0.252 | 0.200 | 0.284 | 0.205 | 0.233 | 0.224 | 0.286 |
| **ETTm1** 96 | 0.189 | 0.262 | **0.270** | **0.325** | 0.288 | 0.339 | 0.279 | 0.333 | 0.285 | 0.344 | 0.290 | 0.341 | 0.300 | 0.353 | 0.293 | 0.345 | 0.289 | 0.342 | 0.314 | 0.367 | 0.300 | 0.345 |
| 192 | 0.218 | 0.284 | **0.310** | **0.352** | 0.326 | 0.368 | 0.320 | 0.358 | 0.331 | 0.368 | 0.337 | 0.369 | 0.341 | 0.380 | 0.335 | 0.372 | 0.329 | 0.368 | 0.374 | 0.410 | 0.336 | 0.366 |
| 336 | 0.242 | 0.300 | **0.343** | **0.373** | 0.363 | 0.394 | 0.348 | 0.377 | 0.359 | 0.386 | 0.374 | 0.393 | 0.374 | 0.396 | 0.368 | 0.386 | 0.362 | 0.390 | 0.413 | 0.432 | 0.367 | 0.387 |
| 720 | 0.249 | 0.296 | **0.400** | **0.405** | 0.417 | 0.419 | 0.405 | 0.408 | 0.415 | 0.416 | 0.437 | 0.428 | 0.429 | 0.430 | 0.426 | 0.417 | 0.416 | 0.423 | 0.753 | 0.613 | 0.419 | 0.417 |
| *Avg.* | 0.225 | 0.286 | **0.331** | **0.364** | 0.349 | 0.380 | 0.338 | 0.369 | 0.348 | 0.380 | 0.359 | 0.382 | 0.361 | 0.390 | 0.356 | 0.380 | 0.349 | 0.381 | 0.464 | 0.456 | 0.356 | 0.379 |
| **ETTm2** 96 | 0.131 | 0.214 | **0.157** | **0.241** | 0.157 | 0.243 | 0.162 | 0.249 | 0.163 | 0.255 | 0.175 | 0.257 | 0.175 | 0.266 | 0.165 | 0.256 | 0.165 | 0.255 | 0.296 | 0.391 | 0.164 | 0.256 |
| 192 | 0.174 | 0.246 | **0.212** | **0.279** | 0.218 | 0.284 | 0.215 | 0.288 | 0.220 | 0.294 | 0.242 | 0.301 | 0.242 | 0.312 | 0.225 | 0.298 | 0.221 | 0.293 | 0.369 | 0.416 | 0.224 | 0.304 |
| 336 | 0.216 | 0.276 | **0.262** | **0.315** | 0.270 | 0.321 | 0.267 | 0.321 | 0.269 | 0.328 | 0.293 | 0.337 | 0.282 | 0.337 | 0.277 | 0.332 | 0.276 | 0.327 | 0.588 | 0.600 | 0.277 | 0.337 |
| 720 | 0.283 | 0.322 | **0.341** | **0.368** | 0.344 | 0.372 | 0.348 | 0.373 | 0.346 | 0.369 | 0.376 | 0.390 | 0.375 | 0.394 | 0.360 | 0.385 | 0.362 | 0.381 | 0.750 | 0.612 | 0.371 | 0.401 |
| *Avg.* | 0.201 | 0.265 | **0.243** | **0.301** | 0.247 | 0.305 | 0.248 | 0.308 | 0.250 | 0.312 | 0.271 | 0.322 | 0.269 | 0.327 | 0.257 | 0.318 | 0.256 | 0.314 | 0.501 | 0.505 | 0.259 | 0.325 |
| **ETTh1** 96 | 0.243 | 0.300 | **0.350** | **0.383** | 0.355 | 0.391 | 0.353 | 0.386 | 0.373 | 0.401 | 0.364 | 0.397 | 0.386 | 0.405 | 0.372 | 0.401 | 0.377 | 0.397 | 0.411 | 0.435 | 0.379 | 0.403 |
| 192 | 0.276 | 0.327 | **0.385** | **0.408** | 0.389 | 0.414 | 0.398 | 0.409 | 0.411 | 0.426 | 0.405 | 0.424 | 0.424 | 0.440 | 0.413 | 0.429 | 0.409 | 0.425 | 0.409 | 0.438 | 0.408 | 0.419 |
| 336 | 0.294 | 0.342 | **0.405** | **0.425** | 0.415 | 0.435 | 0.415 | 0.428 | 0.439 | 0.444 | 0.427 | 0.439 | 0.449 | 0.460 | 0.438 | 0.450 | 0.431 | 0.444 | 0.433 | 0.457 | 0.440 | 440 |
| 720 | 0.266 | 0.322 | **0.422** | **0.449** | 0.443 | 0.462 | 0.436 | 0.458 | 0.495 | 0.493 | 0.439 | 0.459 | 0.495 | 0.487 | 0.486 | 0.484 | 0.457 | 0.477 | 0.501 | 0.514 | 0.471 | 0.493 |
| *Avg.* | 0.270 | 0.323 | **0.391** | **0.416** | 0.401 | 0.426 | 0.401 | 0.420 | 0.430 | 0.441 | 0.409 | 0.430 | 0.439 | 0.448 | 0.427 | 0.441 | 0.419 | 0.436 | 0.439 | 0.461 | 0.425 | 0.439 |
| **ETTh2** 96 | 0.214 | 0.285 | **0.261** | **0.329** | 0.270 | 0.331 | 0.271 | 0.335 | 0.273 | 0.336 | 0.277 | 0.343 | 0.297 | 0.348 | 0.281 | 0.351 | 0.274 | 0.337 | 0.728 | 0.603 | 0.300 | 0.364 |
| 192 | 0.261 | 0.320 | **0.320** | **0.370** | 0.338 | 0.375 | 0.335 | 0.376 | 0.334 | 0.381 | 0.348 | 0.391 | 0.372 | 0.403 | 349 | 0.387 | 0.348 | 0.384 | 0.723 | 0.607 | 0.387 | 0.423 |
| 336 | 0.270 | 0.327 | **0.346** | **0.394** | 0.370 | 0.402 | 0.354 | 0.398 | 0.363 | 0.415 | 0.375 | 0.418 | 0.388 | 0.418 | 0.366 | 0.413 | 0.377 | 0.416 | 0.740 | 0.628 | 0.490 | 0.487 |
| 720 | 0.261 | 0.325 | **0.381** | **0.423** | 0.402 | 0.434 | 0.384 | 0.426 | 0.408 | 0.446 | 0.409 | 0.458 | 0.424 | 0.444 | 0.401 | 0.436 | 0.406 | 0.441 | 1.386 | 0.882 | 0.704 | 0.597 |
| *Avg.* | 0.252 | 0.314 | **0.327** | **0.379** | 0.345 | 0.386 | 0.336 | 0.384 | 0.345 | 0.395 | 0.352 | 0.402 | 0.370 | 0.403 | 0.349 | 0.397 | 0.351 | 0.395 | 0.894 | 0.680 | 0.470 | 0.468 |

Table 6: Full results of long-term forecasting of hyperparameter searching. The Re-Bound column denotes the empirical bound discussed in Sec.3.3. The look-back window is searched from {336, 512, 720} for the best performance. Timer-XL uses a 672-length window as in the original paper. All results are averaged across four different prediction lengths: {96, 192, 336, 720}. The **best** and second-best results are highlighted.

## G  ROUBUSTENESS

In Table 10, we present the error bar of ReNF in datasets with relatively small sizes or high instability. It shows that ReNF exhibits high robustness because of its simple structure, which is a favorable characteristic for industrial applications.

## H  FURTHER EXPLORATIONS

### H.1  TWO FACTORS OF BDO

We wish to clarify that our BDO paradigm is comprised of two distinct and essential mechanisms. The first is the addition of a linear head to each block, which generates an explicit sub-forecast from the intermediate representation. The second is the recursive concatenation of this sub-forecast with the input for the subsequent stage, which encourages the network to implicitly learn a post-combination

| Model | ReNF (Ours) | | OrthoLienar 2025 | | TimeMix. 2024b | | FilterNet 2024 | | FITS 2024 | | DLinear 2023 | | TimeMix.++ 2025 | | Leddam 2024 | | CARD 2024 | | Fredformer 2024 | | iTrans. 2024 | | PatchTST 2023 | |
|---|---|---|---|---|---|---|---|---|---|---|---|---|---|---|---|---|---|---|---|---|---|---|---|---|
| Metric | MSE | MAE | MSE | MAE | MSE | MAE | MSE | MAE | MSE | MAE | MSE | MAE | MSE | MAE | MSE | MAE | MSE | MAE | MSE | MAE | MSE | MAE | MSE | MAE |
| CarSales 24 | **0.263** | **0.283** | 0.320 | 0.302 | 0.320 | 0.318 | 0.318 | 0.319 | 0.359 | 0.347 | 0.354 | 0.350 | 0.323 | 0.348 | 0.325 | 0.322 | 0.337 | 0.321 | 0.319 | 0.326 | 0.303 | 0.312 | 0.319 | 0.319 |
| CarSales 36 | **0.280** | **0.298** | 0.334 | 0.315 | 0.332 | 0.331 | 0.331 | 0.330 | 0.373 | 0.360 | 0.368 | 0.365 | 0.351 | 0.348 | 0.337 | 0.333 | 0.348 | 0.333 | 0.333 | 0.335 | 0.318 | 0.323 | 0.332 | 0.330 |
| CarSales 48 | **0.299** | **0.315** | 0.347 | 0.327 | 0.345 | 0.343 | 0.342 | 0.341 | 0.385 | 0.370 | 0.382 | 0.379 | 0.351 | 0.342 | 0.351 | 0.346 | 0.362 | 0.345 | 0.349 | 0.344 | 0.331 | 0.332 | 0.347 | 0.344 |
| CarSales 60 | **0.315** | **0.328** | 0.358 | 0.337 | 0.355 | 0.351 | 0.352 | 0.349 | 0.399 | 0.385 | 0.388 | 0.380 | 0.363 | 0.352 | 0.361 | 0.353 | 0.372 | 0.353 | 0.359 | 0.349 | 0.344 | 0.342 | 0.355 | 0.348 |
| CarSales Avg | **0.289** | **0.306** | 0.340 | 0.320 | 0.338 | 0.336 | 0.336 | 0.335 | 0.379 | 0.365 | 0.373 | 0.368 | 0.347 | 0.340 | 0.343 | 0.338 | 0.355 | 0.338 | 0.340 | 0.338 | 0.324 | 0.327 | 0.338 | 0.335 |
| Power 24 | **1.268** | **0.855** | 1.343 | 0.870 | 1.341 | 0.881 | 1.410 | 0.916 | 1.491 | 0.944 | 1.390 | 0.916 | 1.340 | 0.877 | 1.397 | 0.909 | 1.406 | 0.886 | 1.410 | 0.913 | 1.462 | 0.924 | 1.468 | 0.935 |
| Power 36 | **1.336** | **0.880** | 1.445 | 0.903 | 1.420 | 0.914 | 1.590 | 0.968 | 1.621 | 0.994 | 1.518 | 0.957 | 1.446 | 0.920 | 1.509 | 0.951 | 1.506 | 0.921 | 1.538 | 0.953 | 1.582 | 0.964 | 1.593 | 0.972 |
| Power 48 | **1.354** | **0.893** | 1.559 | 0.946 | 1.567 | 0.963 | 1.680 | 1.009 | 1.775 | 1.052 | 1.610 | 0.993 | 1.467 | 0.933 | 1.646 | 0.999 | 1.583 | 0.957 | 1.652 | 1.008 | 1.696 | 1.011 | 1.710 | 1.020 |
| Power 60 | **1.383** | **0.913** | 1.602 | 0.971 | 1.609 | 0.988 | 1.776 | 1.053 | 1.958 | 1.122 | 1.679 | 1.020 | 1.626 | 1.006 | 1.727 | 1.043 | 1.693 | 1.003 | 1.752 | 1.049 | 1.796 | 1.061 | 1.829 | 1.064 |
| Power Avg | **1.335** | **0.885** | 1.487 | 0.922 | 1.484 | 0.937 | 1.614 | 0.986 | 1.711 | 1.028 | 1.549 | 0.972 | 1.470 | 0.934 | 1.570 | 0.975 | 1.547 | 0.942 | 1.588 | 0.981 | 1.634 | 0.990 | 1.650 | 0.998 |
| METR-LA 24 | **0.613** | 0.348 | 0.650 | **0.337** | 0.671 | 0.413 | 0.670 | 0.402 | 0.698 | 0.416 | 0.645 | 0.458 | 0.617 | 0.394 | 0.680 | 0.405 | 0.700 | 0.378 | 0.676 | 0.408 | 0.700 | 0.413 | 0.679 | 0.410 |
| METR-LA 36 | **0.748** | 0.398 | 0.800 | **0.388** | 0.841 | 0.480 | 0.824 | 0.471 | 0.874 | 0.490 | 0.785 | 0.533 | 0.781 | 0.457 | 0.841 | 0.471 | 0.874 | 0.448 | 0.852 | 0.477 | 0.867 | 0.480 | 0.845 | 0.484 |
| METR-LA 48 | 0.860 | 0.438 | 0.905 | **0.427** | 0.964 | 0.531 | 0.955 | 0.521 | 1.013 | 0.546 | 0.885 | 0.585 | **0.842** | 0.520 | 0.963 | 0.528 | 1.017 | 0.498 | 0.982 | 0.526 | 1.017 | 0.539 | 0.972 | 0.536 |
| METR-LA 60 | **0.950** | 0.471 | 0.999 | **0.457** | 1.047 | 0.573 | 1.050 | 0.563 | 1.122 | 0.589 | 0.959 | 0.623 | 0.958 | 0.551 | 1.029 | 0.556 | 1.126 | 0.541 | 1.084 | 0.569 | 1.079 | 0.572 | 1.077 | 0.578 |
| METR-LA Avg | **0.793** | 0.414 | 0.838 | **0.402** | 0.881 | 0.499 | 0.875 | 0.489 | 0.927 | 0.510 | 0.819 | 0.550 | 0.799 | 0.480 | 0.878 | 0.490 | 0.929 | 0.466 | 0.898 | 0.495 | 0.916 | 0.501 | 0.893 | 0.502 |
| Website 24 | **0.155** | **0.286** | 0.186 | 0.306 | 0.229 | 0.335 | 0.273 | 0.357 | 0.431 | 0.469 | 0.315 | 0.393 | 0.231 | 0.349 | 0.240 | 0.345 | 0.325 | 0.370 | 0.216 | 0.335 | 0.181 | 0.305 | 0.245 | 0.350 |
| Website 36 | **0.224** | **0.343** | 0.272 | 0.356 | 0.361 | 0.420 | 0.401 | 0.441 | 0.554 | 0.552 | 0.385 | 0.447 | 0.328 | 0.397 | 0.327 | 0.405 | 0.428 | 0.442 | 0.331 | 0.411 | 0.226 | 0.343 | 0.370 | 0.429 |
| Website 48 | 0.283 | 0.386 | 0.365 | 0.391 | 0.501 | 0.507 | 0.530 | 0.522 | 0.694 | 0.647 | 0.436 | 0.486 | 0.450 | 0.473 | 0.446 | 0.475 | 0.457 | 0.478 | 0.483 | 0.496 | **0.263** | **0.370** | 0.504 | 0.513 |
| Website 60 | **0.267** | **0.374** | 0.486 | 0.481 | 0.571 | 0.562 | 0.630 | 0.592 | 0.736 | 0.673 | 0.468 | 0.510 | 0.525 | 0.517 | 0.561 | 0.549 | 0.596 | 0.566 | 0.556 | 0.547 | 0.323 | 0.410 | 0.587 | 0.565 |
| Website Avg | **0.233** | **0.348** | 0.327 | 0.383 | 0.415 | 0.456 | 0.458 | 0.478 | 0.604 | 0.585 | 0.401 | 0.459 | 0.384 | 0.434 | 0.393 | 0.443 | 0.451 | 0.464 | 0.396 | 0.447 | 0.248 | 0.357 | 0.426 | 0.464 |
| SP500 24 | **0.150** | **0.270** | 0.155 | 0.271 | 0.159 | 0.288 | 0.181 | 0.317 | 0.193 | 0.334 | 0.189 | 0.330 | 0.172 | 0.305 | 0.175 | 0.308 | 0.156 | 0.276 | 0.181 | 0.315 | 0.180 | 0.309 | 0.164 | 0.298 |
| SP500 36 | **0.204** | 0.322 | 0.209 | **0.317** | 0.218 | 0.343 | 0.224 | 0.341 | 0.259 | 0.389 | 0.250 | 0.363 | 0.227 | 0.344 | 0.232 | 0.358 | 0.206 | 0.319 | 0.239 | 0.365 | 0.225 | 0.346 | 0.221 | 0.341 |
| SP500 48 | **0.251** | **0.355** | 0.258 | 0.358 | 0.264 | 0.367 | 0.280 | 0.384 | 0.324 | 0.439 | 0.291 | 0.398 | 0.272 | 0.383 | 0.276 | 0.388 | 0.259 | 0.352 | 0.283 | 0.394 | 0.275 | 0.383 | 0.278 | 0.397 |
| SP500 60 | **0.295** | 0.386 | 0.305 | 0.387 | 0.322 | 0.416 | 0.332 | 0.416 | 0.391 | 0.486 | 0.377 | 0.475 | 0.319 | 0.413 | 0.325 | 0.423 | 0.303 | **0.385** | 0.341 | 0.438 | 0.322 | 0.418 | 0.321 | 0.409 |
| SP500 Avg | **0.225** | **0.333** | 0.231 | 0.333 | 0.241 | 0.353 | 0.254 | 0.365 | 0.291 | 0.412 | 0.277 | 0.391 | 0.247 | 0.361 | 0.252 | 0.369 | 0.231 | 0.333 | 0.261 | 0.378 | 0.250 | 0.364 | 0.246 | 0.361 |
| NASDAQ 24 | **0.114** | **0.211** | 0.121 | 0.216 | 0.122 | 0.221 | 0.130 | 0.230 | 0.140 | 0.244 | 0.155 | 0.274 | 0.132 | 0.233 | 0.125 | 0.222 | 0.124 | 0.220 | 0.128 | 0.226 | 0.137 | 0.237 | 0.127 | 0.224 |
| NASDAQ 36 | **0.155** | **0.253** | 0.163 | 0.261 | 0.183 | 0.279 | 0.175 | 0.273 | 0.184 | 0.284 | 0.196 | 0.306 | 0.177 | 0.278 | 0.174 | 0.271 | 0.167 | 0.266 | 0.170 | 0.268 | 0.184 | 0.280 | 0.174 | 0.269 |
| NASDAQ 48 | **0.196** | **0.290** | 0.205 | 0.296 | 0.200 | 0.298 | 0.224 | 0.314 | 0.234 | 0.324 | 0.244 | 0.344 | 0.216 | 0.311 | 0.222 | 0.312 | 0.218 | 0.307 | 0.218 | 0.306 | 0.229 | 0.318 | 0.225 | 0.314 |
| NASDAQ 60 | **0.236** | **0.321** | 0.259 | 0.336 | 0.238 | 0.328 | 0.259 | 0.340 | 0.282 | 0.357 | 0.318 | 0.401 | 0.249 | 0.337 | 0.264 | 0.341 | 0.264 | 0.341 | 0.262 | 0.339 | 0.279 | 0.352 | 0.265 | 0.339 |
| NASDAQ Avg | **0.175** | **0.269** | 0.187 | 0.277 | 0.186 | 0.281 | 0.197 | 0.289 | 0.210 | 0.302 | 0.228 | 0.331 | 0.193 | 0.290 | 0.196 | 0.286 | 0.193 | 0.284 | 0.194 | 0.285 | 0.207 | 0.297 | 0.198 | 0.286 |

Table 7: Full results for the short-term forecasting. We use the look-back window with length $T = 36$ to predict lengths $\{24, 36, 48, 60\}$. The **best results** and second-best results are highlighted.

function as motivated by MNFP. To disentangle their respective contributions, we conduct an ablation study on these two factors.

The results on the Electricity, ETTh1, and ETTm1 datasets, shown in Table 11, indicate that the two components are synergistic; removing either one leads to a notable degradation in performance. Overall, we find that the hierarchical supervision from the sub-forecasts plays a more significant role. This is expected, as it not only allows for repeated reuse of label information but also enforces a causal and homogeneous structure on the predictive representations across layers, an effect we visualize in Figure 12. However, a few anomalous results suggest that other unresolved factors, such as the quality of the ground truth labels, may also influence performance in certain cases.

## H.2    PREFERENCE OF DEEP REPRESENTATIONS.

In Sec. 2.4, we introduce two variants of MLPs with a few differences. The primary distinction is the inclusion of skip-connections between representations in ReNF-$\beta$, which facilitates the learning of deeper, more complex non-linear dynamics. To justify this design choice, the following experiments demonstrate the performance degradation that occurs when a model's complexity is mismatched with the dataset's intrinsic characteristics.

Specifically, we apply the ReNF-$\beta$ to the ETTh1 and ETTh2 datasets, and apply the alpha version to the large volume Electricity and Traffic datasets. The results are shown in the Table 12, from which we can deduce that the complex deep representations of ReNF-$\beta$ are clearly detrimental to ETTh datasets. In stark contrast, large-volume datasets like Electricity and Traffic benefit from deeper, more expressive representations. This finding provides a direct explanation, from the perspective of representation depth, for the recurring phenomenon where simpler, parsimonious NFs outperform more complex ones on certain benchmarks. It underscores the critical importance of matching model capacity to the intrinsic characteristics of the time series data, identifying a clear direction for future work in adaptive forecaster design.

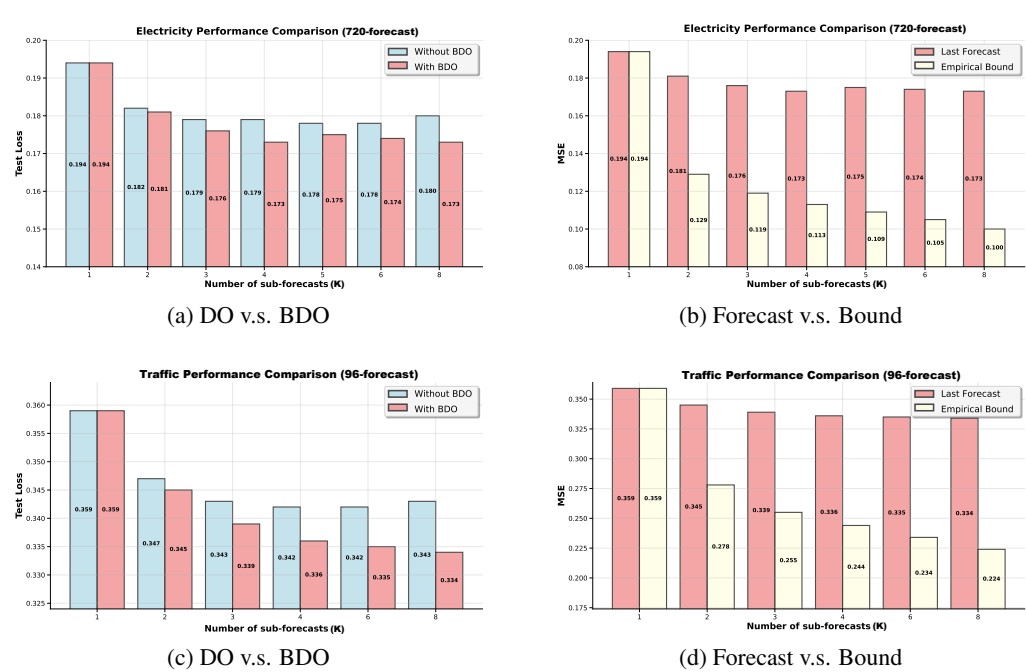

Figure 11: Supplementary illustrations of the performance variation with different numbers of sub-forecasts K. (a). Comparison of using DO or BDO with the Electricity dataset. (b).Comparison between the best forecast and the empirical bound of ReNF with the Electricity dataset. (c). Comparison of using DO or BDO with the Traffic dataset. (d).Comparison between the best forecast and the empirical bound of ReNF with the Traffic dataset.

| Variants | | ReNF origin | | ReNF w/o EMA | |
|---|---|---|---|---|---|
| Metric | | MSE | MAE | MSE | MAE |
| ETTh1 | 96 | 0.350 | 0.383 | 0.357 | 0.387 |
| | 192 | 0.385 | 0.408 | 0.407 | 0.421 |
| | 336 | 0.405 | 0.425 | 0.426 | 0.435 |
| | 720 | 0.422 | 0.449 | 0.447 | 0.467 |
| | *Avg.* | 0.391 | 0.416 | 0.409 | 0.428 |
| ETTh2 | 96 | 0.261 | 0.329 | 0.265 | 0.329 |
| | 192 | 0.320 | 0.370 | 0.336 | 0.375 |
| | 336 | 0.346 | 0.394 | 0.397 | 0.418 |
| | 720 | 0.381 | 0.423 | 0.421 | 0.441 |
| | *Avg.* | 0.327 | 0.379 | 0.355 | 0.391 |
| ETTm1 | 96 | 0.270 | 0.325 | 0.303 | 0.347 |
| | 192 | 0.310 | 0.352 | 0.341 | 0.371 |
| | 336 | 0.343 | 0.373 | 0.362 | 0.386 |
| | 720 | 0.401 | 0.406 | 0.425 | 0.420 |
| | *Avg.* | 0.331 | 0.364 | 0.358 | 0.381 |
| ETTm2 | 96 | 0.157 | 0.241 | 0.162 | 0.246 |
| | 192 | 0.212 | 0.279 | 0.221 | 0.285 |
| | 336 | 0.262 | 0.315 | 0.272 | 0.322 |
| | 720 | 0.341 | 0.368 | 0.352 | 0.374 |
| | *Avg.* | 0.243 | 0.301 | 0.252 | 0.307 |

| Variants | | ReNF origin | | ReNF w/o EMA | |
|---|---|---|---|---|---|
| Metric | | MSE | MAE | MSE | MAE |
| Weather | 96 | 0.138 | 0.180 | 0.139 | 0.182 |
| | 192 | 0.181 | 0.224 | 0.184 | 0.227 |
| | 336 | 0.231 | 0.266 | 0.232 | 0.266 |
| | 720 | 0.304 | 0.318 | 0.308 | 0.320 |
| | *Avg.* | 0.214 | 0.247 | 0.216 | 0.249 |
| Electricity | 96 | 0.118 | 0.210 | 0.124 | 0.218 |
| | 192 | 0.138 | 0.229 | 0.145 | 0.239 |
| | 336 | 0.151 | 0.244 | 0.156 | 0.252 |
| | 720 | 0.173 | 0.266 | 0.182 | 0.277 |
| | *Avg.* | 0.145 | 0.237 | 0.152 | 0.247 |
| Traffic | 96 | 0.335 | 0.226 | 0.341 | 0.237 |
| | 192 | 0.356 | 0.239 | 0.363 | 0.249 |
| | 336 | 0.366 | 0.246 | 0.373 | 0.256 |
| | 720 | 0.402 | 0.267 | 0.411 | 0.278 |
| | *Avg.* | 0.365 | 0.245 | 0.372 | 0.255 |
| Solar | 96 | 0.157 | 0.202 | 0.177 | 0.232 |
| | 192 | 0.174 | 0.210 | 0.193 | 0.234 |
| | 336 | 0.180 | 0.219 | 0.195 | 0.239 |
| | 720 | 0.190 | 0.225 | 0.199 | 0.242 |
| | *Avg.* | 0.176 | 0.214 | 0.191 | 0.237 |

Table 8: Full numerical results on the effect of EMA smoothing.

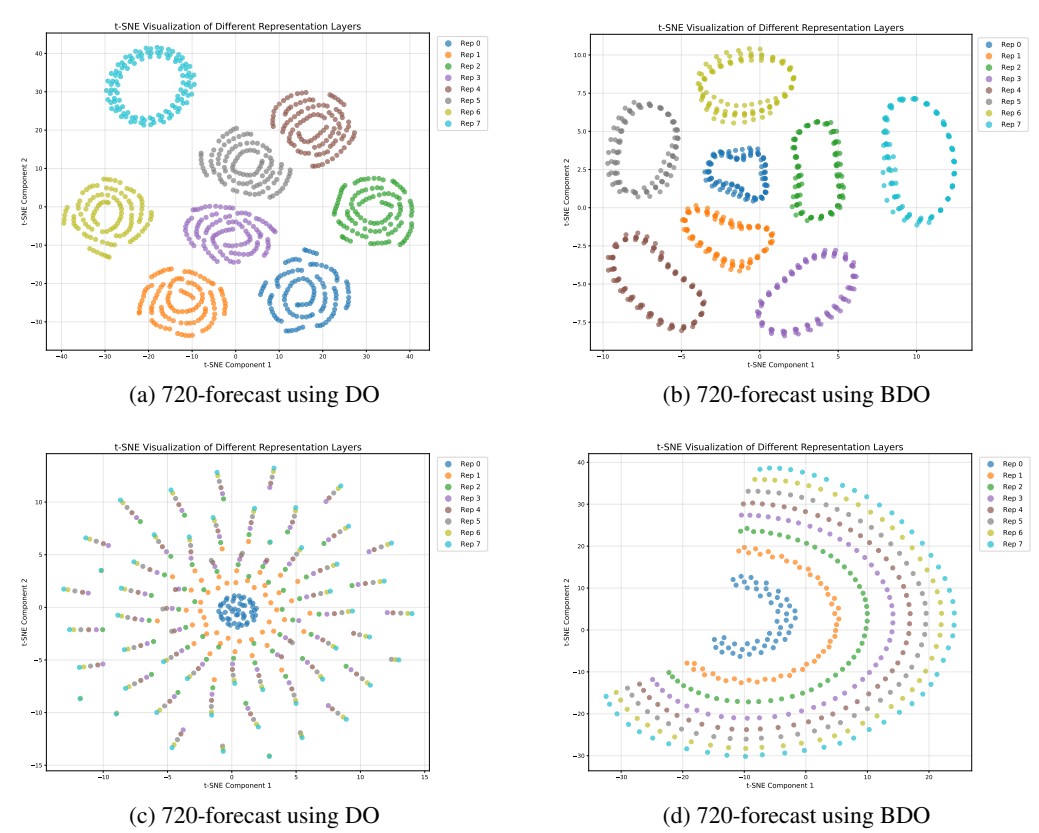

(a) 720-forecast using DO

(b) 720-forecast using BDO

(c) 720-forecast using DO

(d) 720-forecast using BDO

Figure 12: Visualizations of the representations of different layers/sub-forecasters. (a). representations of using DO with the ETTh1 dataset. (b). representations of using BDO with the ETTh1 dataset. (c). representations of using DO with the ETTm1 dataset. (d). representations of using BDO with the ETTm1 dataset. This indicates that BDO tends to form predictive representations that are homogeneous and hierachical.

| Variants | | ETTh1 | | | | | ETTh2 | | | | | Electricity | | | | |
|---|---|---|---|---|---|---|---|---|---|---|---|---|---|---|---|---|
| Metric | | 96 | 192 | 336 | 720 | Avg. | 96 | 192 | 336 | 720 | Avg. | 96 | 192 | 336 | 720 | Avg. |
| ReNF | MSE | 0.350 | 0.385 | 0.405 | 0.422 | 0.391 | 0.261 | 0.320 | 0.346 | 0.381 | 0.327 | 0.118 | 0.138 | 0.151 | 0.173 | 0.145 |
| | MAE | 0.383 | 0.408 | 0.425 | 0.449 | 0.416 | 0.329 | 0.370 | 0.394 | 0.423 | 0.379 | 0.210 | 0.229 | 0.244 | 0.266 | 0.237 |
| ReNF w/o pre-drop | MSE | 0.352 | 0.385 | 0.406 | 0.428 | 0.393 | 0.260 | 0.321 | 0.348 | 0.383 | 0.328 | 0.119 | 0.138 | 0.150 | 0.175 | 0.146 |
| | MAE | 0.386 | 0.410 | 0.429 | 0.454 | 0.420 | 0.329 | 0.370 | 0.396 | 0.425 | 0.380 | 0.210 | 0.229 | 0.243 | 0.267 | 0.237 |

Table 9: Effects of pre-drop.

| Model | ReNF | | TimeBridge | | Confidence |
|---|---|---|---|---|---|
| Dataset | MSE | MAE | MSE | MAE | Interval |
| ETTh1 | $0.391 \pm 0.000$ | $0.417 \pm 0.000$ | $0.399 \pm 0.009$ | $0.424 \pm 0.008$ | 99% |
| ETTh2 | $0.327 \pm 0.000$ | $0.380 \pm 0.000$ | $0.343 \pm 0.018$ | $0.383 \pm 0.014$ | 99% |
| Weather | $0.214 \pm 0.000$ | $0.247 \pm 0.000$ | $0.219 \pm 0.006$ | $0.250 \pm 0.003$ | 99% |
| Solar | $0.177 \pm 0.000$ | $0.215 \pm 0.000$ | $0.182 \pm 0.003$ | $0.219 \pm 0.003$ | 99% |

Table 10: Standard deviation and statistical tests for ReNF and TimeBridge on ETTh1, ETTh2, Weather, and Solar datasets. The results are based on the average performance across four prediction lengths from five runs with different random seeds.

| Variants | | ReNF origin | | ReNF w/o factor_1 | | ReNF w/o factor_2 | |
|---|---|---|---|---|---|---|---|
| Metric | | MSE | MAE | MSE | MAE | MSE | MAE |
| Electricity | 96 | 0.118 | 0.210 | 0.119 | 0.211 | 0.123 | 0.215 |
| | 192 | 0.138 | 0.229 | 0.138 | 0.229 | 0.145 | 0.236 |
| | 336 | 0.151 | 0.244 | 0.153 | 0.245 | 0.156 | 0.251 |
| | 720 | 0.173 | 0.266 | 0.179 | 0.271 | 0.177 | 0.269 |
| | Avg. | 0.145 | 0.237 | 0.147 | 0.239 | 0.150 | 0.243 |
| ETTh1 | 96 | 0.350 | 0.383 | 0.356 | 0.382 | 0.356 | 0.382 |
| | 192 | 0.385 | 0.408 | 0.394 | 0.406 | 0.394 | 0.407 |
| | 336 | 0.405 | 0.427 | 0.420 | 0.427 | 0.420 | 0.425 |
| | 720 | 0.422 | 0.449 | 0.432 | 0.454 | 0.436 | 0.456 |
| | Avg. | 0.391 | 0.417 | 0.400 | 0.417 | 0.402 | 0.418 |
| ETTm1 | 96 | 0.270 | 0.325 | 0.273 | 0.327 | 0.275 | 0.327 |
| | 192 | 0.310 | 0.352 | 0.311 | 0.355 | 0.312 | 0.353 |
| | 336 | 0.343 | 0.373 | 0.343 | 0.376 | 0.342 | 0.374 |
| | 720 | 0.401 | 0.406 | 0.398 | 0.409 | 0.413 | 0.410 |
| | Avg. | 0.331 | 0.364 | 0.331 | 0.367 | 0.336 | 0.366 |
| Solar | 96 | 0.157 | 0.202 | 0.165 | 0.201 | 0.170 | 0.217 |
| | 192 | 0.174 | 0.210 | 0.185 | 0.215 | 0.195 | 0.226 |
| | 336 | 0.180 | 0.219 | 0.185 | 0.219 | 0.189 | 0.233 |
| | 720 | 0.190 | 0.225 | 0.197 | 0.227 | 0.197 | 0.238 |
| | Avg. | 0.176 | 0.214 | 0.183 | 0.216 | 0.188 | 0.229 |

Table 11: Ablations on the two factors of our BDO implementation. The factor_1 denotes the concatenation in input space, and factor_2 denotes the computation of the sub-forecasting losses.

| Variants | | ReNF origin | | ReNF exchange | |
|---|---|---|---|---|---|
| Metric | | MSE | MAE | MSE | MAE |
| ETTh1 | 96 | 0.350 | 0.383 | 0.384 | 0.410 |
| | 192 | 0.385 | 0.408 | 0.407 | 0.428 |
| | 336 | 0.405 | 0.425 | 0.447 | 0.456 |
| | 720 | 0.422 | 0.449 | 0.502 | 0.493 |
| | Avg. | 0.391 | 0.416 | 0.435 | 0.447 |
| ETTh2 | 96 | 0.261 | 0.329 | 0.271 | 0.332 |
| | 192 | 0.320 | 0.370 | 0.340 | 0.376 |
| | 336 | 0.346 | 0.394 | 0.381 | 0.410 |
| | 720 | 0.381 | 0.423 | 0.415 | 0.437 |
| | Avg. | 0.327 | 0.379 | 0.352 | 0.389 |
| Electricity | 96 | 0.118 | 0.210 | 0.125 | 0.217 |
| | 192 | 0.138 | 0.229 | 0.144 | 0.235 |
| | 336 | 0.151 | 0.244 | 0.160 | 0.252 |
| | 720 | 0.173 | 0.266 | 0.196 | 0.283 |
| | Avg. | 0.145 | 0.237 | 0.156 | 0.247 |
| Traffic | 96 | 0.335 | 0.226 | 0.359 | 0.241 |
| | 192 | 0.356 | 0.239 | 0.377 | 0.249 |
| | 336 | 0.366 | 0.246 | 0.389 | 0.259 |
| | 720 | 0.402 | 0.267 | 0.426 | 0.284 |
| | Avg. | 0.365 | 0.245 | 0.388 | 0.258 |

Table 12: Performances degrade drastically after using NFs with an improper degree of complexity. The Exchange column denotes that the version of ReNF ($\alpha$ and $\beta$) is alternated.

# I VISUALIZATION OF PREDICTION

In the following, we present the visualizations of multivariate long-term time series forecasting using ReNF. The predicted variable in each figure is randomly selected.

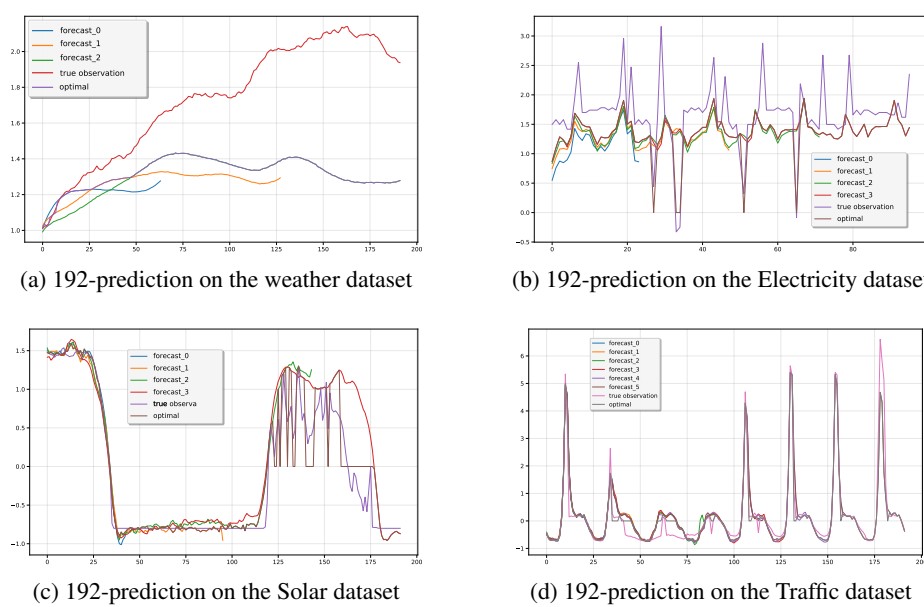

(a) 192-prediction on the weather dataset

(b) 192-prediction on the Electricity dataset

(c) 192-prediction on the Solar dataset

(d) 192-prediction on the Traffic dataset

Figure 13: Visualization of forecasting results of ReNF. The figure shows multiple outputs of ReNF in different layers, along with the result of applying optimal post-combination.

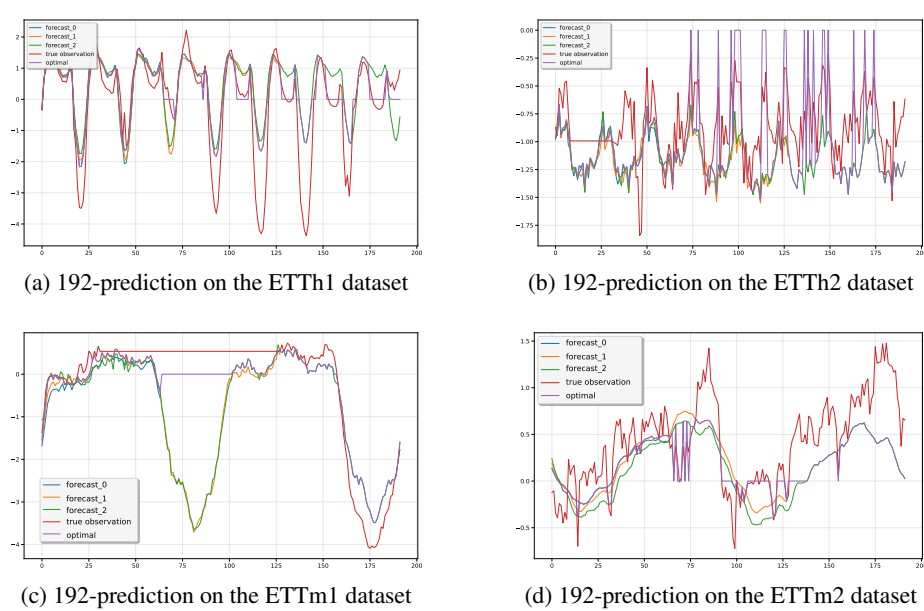

(a) 192-prediction on the ETTh1 dataset

(b) 192-prediction on the ETTh2 dataset

(c) 192-prediction on the ETTm1 dataset

(d) 192-prediction on the ETTm2 dataset

Figure 14: Visualization of forecasting results of ReNF. The figure shows multiple outputs of ReNF in different layers, along with the result of applying optimal post-combination.

## J LIMITATION AND FURTHER DISCUSSION

The Multiple Neural Forecasting Proposition (MNFP) presented in this paper is foundational but preliminary. While it provides the core intuition for our work, a more rigorous and deeper exploration of its theoretical properties or deriving other related theorems could inspire new and promising research directions.

Specifically, under the paradigm of MNFP, the role of Neural Network (NN) in this area becomes transparent. First and perhaps most importantly, we should leverage NN's capability to create a powerful post-combination function. While our BDO paradigm is effective, the current recursive strategy for combining sub-forecasts is not fully optimal. This is evident from the performance gap between our final forecast and the theoretical empirical bound as the number of stages increases. Developing more intricate methods to better leverage the full set of sub-forecasts could lead to substantial accuracy gains. Furthermore, the benefit of BDO is less pronounced on certain datasets, such as ETTm2. The underlying reasons for this variance warrant further investigation.

Second, we are consistently supposed to build more powerful Neural Forecasting Machines (NFMs) to approximate the expected distributions of future data, thereby reducing the bias $b$ in the MNFP C. Therefore, a comprehensive study is needed to verify the effects of our proposed techniques when applied to other advanced model architectures beyond MLPs.

Finally, this work develops the BDO paradigm specifically for the LTSF setting. A key open question is how this paradigm can be better adapted for diverse forecasting tasks with short input, which could ultimately lead to a more unified framework for time series forecasting.

## K DECLARATION OF THE LLMS USE

We utilized a Large Language Model (LLM) to assist in refining the language of this manuscript. The LLM was used specifically to check for grammatical correctness and to improve the clarity and flow of expressions, ensuring a professional academic tone. The sole purpose of using the LLM was to enhance the readability and comprehensibility of the paper based on our first draft. All other contents, including the core ideas, presentation logic, experimental design, and results, are entirely our own.

