# OpenReview forum: "ReNF: Rethinking the Design Space of Neural Long-Term Time Series Forecasters"
_ICLR.cc/2026/Conference — ICLR 2026 Conference Withdrawn Submission_

### Official Review · Reviewer_ApEN · 2025-10-29

**Soundness:** 3
**Presentation:** 3
**Contribution:** 3
**Rating:** 6
**Confidence:** 4

**Summary:**

This paper proposes ReNF, a simple MLP-based model for long-term time series forecasting (LTSF). It introduces the Multiple Neural Forecasting Theorem (MNFT) and a Boosted Direct Output (BDO) strategy that combines auto-regressive and direct forecasting benefits. ReNF achieves state-of-the-art performance on major benchmarks with fewer parameters, showing that strong forecasting principles can outperform complex architectures.

**Strengths:**

1. The motivation is clear, it aims to simplify the problem while providing solid theoretical justification.
2. The experiments are comprehensive, with representative baselines and strong overall performance.

**Weaknesses:**

1. Theorem 1 claims that an accurate forecast is theoretically achievable even with a rather weak generator. Could the authors clarify how this insight is formally derived from the theorem’s assumptions?

2. Why would combining multiple methods yield more accurate results than any single candidate? Please provide either theoretical or empirical justification.

3. The paper introduces a new training strategy, BDO, but this raises concerns about computational cost. Does the “Efficiency” metric in Table 1 refer to training or inference (testing) time?

4. The paper lacks results averaged over multiple random seeds. Including the standard deviation would make the evaluation more statistically reliable.

**Questions:**

see weaknesses

---

> ### Author Response · Authors · 2025-11-18
> **Respond to reviewer ApEN (1).**
>
> Dear Reviewer ApEN,
>
> We appreciate your comprehension and the key support for our work. Here are our responses to your questions one by one,
>
> **W1: About the intuition of Theorem 1**
>
> >Thank you for this question. It allows us to clarify the core intuition behind our proposition, which is best understood as having two complementary parts.
>
> >**1. The Power of Multiple Forecasts (c):**
> >The central intuition is a formalization of the "wisdom of the crowds" principle from ensemble forecasting. The key term in our derived bound is c, the number of candidate forecasts. The proposition makes it explicit that the error bound can be systematically reduced by increasing c.
>
> >This is because averaging a large number of diverse forecasts helps to cancel out random, non-systematic estimation errors—a principle related to the Law of Large Numbers. In an extreme (and impractical) scenario, even a weak generator could approach an accurate prediction with a sufficient number of samples. This highlights the powerful, yet often overlooked, role of generating multiple forecasts.
>
> >**2. The Indispensable Role of a Powerful NFM (to reduce b):**
> >However, this "brute force" approach has a fundamental limit, which brings us to the second key intuition. Our proposition includes an irreducible **predictive bias term, b**, which represents the systematic error of the base forecaster. **This bias cannot be reduced by averaging, no matter how large c is.**
>
> >This is precisely why the pursuit of more powerful NFMs remains essential. A "stronger" NFM is one that is better able to learn the true data-generating process, thereby achieving a lower predictive bias b.
>
> >The proposition provides a complete picture. We need **many forecasts (c)** to tackle the error from *variance*, and a **powerful NFM** to tackle the error from *bias*. Our work's primary contribution is to highlight and operationalize the first part—leveraging c through the BDO paradigm—which has been largely overlooked in the recent pursuit of complex, monolithic architectures. We have also provided further details on this in our responses to Reviewers PbGb and kCsT.
> ****
> **W2:  Why would combining multiple methods yield more accurate results than any single candidate.**
>
> >Thank you for this question, which addresses a foundational principle that is central to our work. The concept that combining multiple forecasts can yield a more accurate result than any single candidate is well-established in the literature on **forecast combination and ensemble methods [1]**.
>
> >Our work builds upon this principle in two key ways:
>
> >**1. Theoretical Formalization:**
> >Our **Proposition 1** provides a formal, theoretical basis for this idea in the context of neural forecasters. It derives an error bound that explicitly includes the number of forecasts, c, demonstrating that the total error bound decreases as c increases. This is because averaging multiple diverse forecasts helps to cancel out non-systematic variance in the predictions.
>
> >**2. Empirical Validation:**
> >Crucially, we provide direct, empirical proof of this principle within our own framework. As demonstrated in **Section 3.3**, the performance of the "oracle" combination—our empirical bound—consistently and significantly improves as we increase the number of sub-forecasts generated by the BDO paradigm.
>
> >This provides clear evidence that the set of hierarchical forecasts from our model contains complementary information, and that an effective combination of these forecasts is fundamentally more powerful than relying on any single one alone.
> ****
> [1] Robert T Clemen. Combining forecasts: A review and annotated bibliography. International journal
> of forecasting, 5(4):559–583, 1989.

---

> > ### Author Response · Authors · 2025-11-18
> > **Respond to reviewer ApEN (2).**
> >
> > **W4:  Including the standard deviation would make the evaluation more statistically reliable**
> >
> > >Thank you for your reminder. We completely agree that including standard deviations is crucial for a statistically reliable evaluation.
> >
> > >We are happy to confirm that we have already conducted this analysis to assess our model's stability. You can find the results in the appendix in our **Robustness section (Table 10)**.
> >
> > >The table reports the standard deviation and confidence intervals over multiple runs with different random seeds. The results demonstrate that ReNF exhibits a very low standard deviation, confirming its high robustness. This result aligns with our expectations, as the simplicity of the underlying MLP architecture naturally lends itself to more stable and reproducible training outcomes.
> > ****

---

> ### Author Response · Authors · 2025-11-18
>
> Dear Reviewer ApEN,
>
> We thank you again for your effort.
> In the revised version, we have substantially improved both the theoretical analysis and the experimental validation. Specifically, we have corrected and clarified the theoretical framework and expanded the appendix with further analysis to strengthen the motivation for our proposals.
>
> Furthermore, we have broadened the empirical scope by evaluating our method on a short-term forecasting task with more diverse datasets and an expanded set of baselines. We are confident that these comprehensive improvements have significantly enhanced the quality and contribution of our paper.
>
> Best,
>
> Authors

---

### Official Review · Reviewer_kwgb · 2025-11-02

**Soundness:** 3
**Presentation:** 3
**Contribution:** 3
**Rating:** 6
**Confidence:** 4

**Summary:**

In this paper, authors provide a new framework based on neural forecasters. Based on the new introduced multiple neural forecasting theorem, it provides both theoretical and experimental support for the effectiveness of this model.

**Strengths:**

1. The comprehensiveness of Multiple Neural Forecasting Theorem and experimental discussion.
2. This model has a better performance than many state-of-art baselines.
3. There is a relatively comprehensive discussion about the EMA and BDO in the ablation study section.

**Weaknesses:**

1. Can authors provide an explanation about the existence of empirical bound? Can we use more advanced model to have a much better performance, or this bound is unsolvable?
2. Can this model combine with other TSF structures, such as transformer?
3. Why adopted both deterministic and frequency based loss for this model? Are there any theoretical support?

**Questions:**

See weaknesses.

---

> ### Author Response · Authors · 2025-11-18
> **Respond to reviewer kwgb.**
>
> Dear Reviewer kwgb,
>
> We sincerely appreciate your review of our work and your key support. Here are our responses to your questions one by one,
> **W1: Existence of the empirical bound**
>
> >Thank you for this interesting question. The empirical bound is a direct and intentional consequence of our BDO paradigm's multi-output nature.
>
> **1. How the Bound is Derived:**
> >Since BDO generates a distinct sub-forecast at each block, it provides a natural set of candidate predictions for the same future horizon. As described in **Section 3.3**, this bound is calculated using an "oracle" post-combination strategy:
>
> - For each individual timestep in the forecast horizon, we select the single best prediction from the set of all candidate forecasts (i.e., the one with the smallest error relative to the ground truth).
>
> >This makes the bound a theoretical upper limit on performance for our model, as it requires knowledge of the future and is therefore unattainable in practice.
>
> **2. The Significance of the Bound:**
> >The significance of this bound is not its attainability, but the insight provided by the **large gap** between it and our model's final forecast. This gap serves as powerful empirical evidence for our central thesis:
>
> - A substantial amount of predictive potential is locked within the model's own forecasts.
> - The key to unlocking this potential lies in learning a more effective **combination function**—a task for which our recursive BDO paradigm is explicitly designed.
>
> >This also highlights a promising direction for future work. Applying our BDO paradigm to more powerful base architectures could not only improve the quality of the individual sub-forecasts but also enrich their diversity, providing an even stronger set of candidates for the implicit combination process to leverage.
> ****
> **W2:  Can this model combine with other TSF structures**
> >Yes. The BDO paradigm is architecturally agnostic and was intentionally designed for simplicity and modularity, making it broadly applicable to other TSF structures.
>
> >Its core mechanisms can be integrated into virtually any deep neural network:
>
> 1. **Hierarchical Supervision:** Linear heads can be added to any intermediate layer to generate sub-forecasts and apply our multi-stage loss.
> 2. **Recursive Combination:** These sub-forecasts can then be flexibly concatenated with the input to the next block, encouraging the model to learn a combination function.
>
> >Our own implementation, detailed in Section 2.4, serves as a template, and our ablation studies (**Section H.1 in the revised version**) confirm the synergistic value of both mechanisms.
>
> >As we state in our "Limitations and Future Work" section, we believe applying BDO to other advanced architectures is a highly promising direction for future research. Given its ensemble-like nature, the BDO paradigm should be able to effectively leverage and enhance the capabilities of more powerful base forecasters.
>
> ****
> **W3:  Why adopted both deterministic and frequency based loss for this model.**
> > As indicated in the first paragraph in section 2.5. This hybrid loss was proposed in the excellent work [], which evidenced the significant effects of reducing the autocorrelation in the label via including a frequency loss. They have analysed this function both theoretically and empirically. For more details, please refer to the above paper.
>
> [1] Wang, Hao, et al. "FreDF: Learning to Forecast in the Frequency Domain." The Thirteenth International Conference on Learning Representations.

---

> ### Author Response · Authors · 2025-11-18
>
> Dear Reviewer kwgb,
>
> We thank you again for your effort.
> In the revised version, we have substantially improved both the theoretical analysis and the experimental validation. Specifically, we have corrected and clarified the theoretical framework and expanded the appendix with further analysis to strengthen the motivation for our proposals.
>
> Furthermore, we have broadened the empirical scope by evaluating our method on a short-term forecasting task with more diverse datasets and an expanded set of baselines. We are confident that these comprehensive improvements have significantly enhanced the quality and contribution of our paper.
>
> Best,
>
> Authors

---

### Official Review · Reviewer_kCsT · 2025-11-03

**Soundness:** 1
**Presentation:** 1
**Contribution:** 2
**Rating:** 2
**Confidence:** 4

**Summary:**

The paper targets long-term time-series forecasting (LTSF), arguing that progress has over-focused on architecture tricks while under-using supervision and stable training. It introduces a Multiple Neural Forecasting Theorem (MNFT): with multiple forecast candidates from a neural forecaster, an (oracle) combination can bound error. A crude upper bound scales with horizon, data range, and number of candidates. Assumptions include bounded data, multiple candidates, independent output elements, and bounded bias. Paper proposes Boosted Direct Output (BDO): recursively forecast longer segments by concatenating previous (shorter-horizon) predictions to the inpu, mixing AR-like causality with DO’s low error propagation. Trains an MLP with a stage-wise, time+frequency loss, heavier weight on earlier (shorter) stages. Uses EMA "shadow" weights for evaluation to smooth validation/test curves and avoid unstable early stopping.

**Strengths:**

- Reproducibility checklist and code provided

- Theoretical results are presented

- BDO recipe that adds causal structure to DO without heavy architecture

**Weaknesses:**

- The focus on long horizon problems undermines the claims of approach's generality

- Theoretical assumptions A1-A4 are not motivated and the significance of Theorem 1 is unclear

- BDO strategy, which seems to be one of core contributions seems trivial, if it is not accompanied with additional block-level supervision. The latter is not described in the paper in any level of detail at all.

- EMA is not novel

**Questions:**

> Furthermore, the field’s focus has shifted towards designing specialized modules for properties that
are believed to be beneficial for LTSF, such as multi-scale (Wang et al., 2024b) and non-stationary
(Liu et al., 2022). However, the progress has become erratic because these architectural additions
often yield subtle gains while overlooking fundamental principles.

- I actually like this reflection a lot. However, I feel that the authors fall in the same trap as many previous works: by focusing on a subset of problems represented only by long-horizon benchmarks, authors fail to answer the question about generality of their approach and its applicability to wide array of problems. Could you please provide additional results on M4, M3 and TOURISM benchmarks? I have observed in the past that the works focusing on long-horizon benchmarks fail to deliver any tangible gains on other benchmarks, resonating with the authors' original sentiment.

- The practical applicability of theoretical assumptions A1-A4 is not clear. Could you please provide a discussion for each, linking it to the practice of forecasting? I am especially interested in A3, how does this work in practice? For example, if we generate each sequence from the same architecture weights, how can each element in an output series be independent?


- Can you explain the significance of Theorem 1? The only takeaway I get from it is that no matter how many candidates $c$ are generated, there is an irreducible error bound related to $b$ and $\sigma_t$, which is arbitrarily loose. In my view, this is trivial and does not provide insight on how the proposed MNFT can improve on top existing approaches.

> Furthermore, higher-quality historical data and a more powerful NFM (which reduces the estimation
error b) both serve to constrain the solution space, leading to better predictions.

- I do not understand how a more powerful NFM should necessarily lead to the reduction of estimation
error $b$. What is the mathematical principle here and how is this related to overfitting? This seems like a rather hand-wavy argument to me.

- I do not understand the significance of Figure 2 and I do not believe it is referenced/discussed in text.

- BDO strategy, which seems to consist in combining the multi-horizon forecast with AR forecast by feeding generated chunks back into the input is a standard practice in today forecasting. For example, Chronos does it. Could you please explain the novelty in more detail? For example, the key contribution behind BDO seems to be the supervision of each output $k$ (e.g. Figure 4), but this supervision scheme is not described mathematically in Section 2.5. Could you please provide the exact formula for BDO loss? Additionally, the BDO supervision is not ablated. What happens if block level supervision is replaced with the supervision of the final output only?

- EMA of model weights is not novel. It is a well-established training/evaluation trick (a.k.a. Polyak/EMA averaging) used for decades and popularized across modern deep learning (e.g. attention is all you need, diffusion models, and many vision/NLP training recipes). Using EMA can absolutely help stability and accuracy, but it should not be positioned as a core contribution. Also, when comparing to other models as baselines, those models should be trained with EMA as well.

---

> ### Author Response · Authors · 2025-11-18
> **Respond to Reviewer kCsT (1)**
>
> Dear Reviewer kCsT,
>
> We sincerely appreciate your efforts in reviewing our paper, and below are our responses to your concerns.
> **W1:  focus on long horizon problems undermines the claims of approach's generality **
>
> > Thank you for this insightful question regarding the generality of our approach. We agree that evaluating our method beyond its primary design focus is essential.
>
> > Our BDO paradigm was indeed designed primarily to tackle the unique challenges of **Long-Term Time Series Forecasting (LTSF)**, such as error accumulation, and its benefits are therefore most pronounced in that setting.
>
> > However, to directly address your concern about generality, we have conducted a new suite of experiments on **short-term forecasting tasks**. For this, we selected datasets with diverse characteristics from a Kaggle competition benchmark. To ensure a fair and objective comparison, we adopted the experimental protocol from the OLinear paper [1] and used their reported SOTA results for our baselines.
>
> > The results, presented below, demonstrate that ReNF achieves highly competitive, and often state-of-the-art, performance even in this short-term setting.
>
> **Average Results for Short-Term Forecasting (MSE / MAE)**
> | Model           | CarSales          | Power             | METR-LA             | Website           | SP500               | NASDAQ            |
> | --------------- | ----------------- | ----------------- | ------------------- | ----------------- | ------------------- | ----------------- |
> | **ReNF (Ours)** | **0.289 / 0.306** | **1.335 / 0.885** | **0.793** / *0.414* | **0.233 / 0.348** | **0.225 / 0.333**   | **0.175 / 0.269** |
> | OrthoLienar     | 0.340 / *0.320*   | 1.487 / *0.922*   | 0.838 / **0.402**   | 0.327 / 0.383     | *0.231* / **0.333** | 0.187 / 0.277     |
> | TimeMixer       | 0.338 / 0.336     | 1.484 / 0.937     | 0.881 / 0.499       | 0.415 / 0.456     | 0.241 / 0.353       | *0.186 / 0.281*   |
> | FilterNet       | 0.336 / 0.335     | 1.614 / 0.986     | 0.875 / 0.489       | 0.458 / 0.478     | 0.254 / 0.365       | 0.197 / 0.289     |
> | FITS            | 0.379 / 0.365     | 1.711 / 1.028     | 0.927 / 0.510       | 0.604 / 0.585     | 0.291 / 0.412       | 0.210 / 0.302     |
> | DLinear         | 0.373 / 0.368     | 1.549 / 0.972     | 0.819 / 0.550       | 0.401 / 0.459     | 0.277 / 0.391       | 0.228 / 0.331     |
> | TimeMixer++     | 0.347 / 0.340     | *1.470* / 0.934   | *0.799* / 0.480     | 0.384 / 0.434     | 0.247 / 0.361       | 0.193 / 0.290     |
> | Leddam          | 0.343 / 0.338     | 1.570 / 0.975     | 0.878 / 0.490       | 0.393 / 0.443     | 0.252 / 0.369       | 0.196 / 0.286     |
> | CARD            | 0.355 / 0.338     | 1.547 / 0.942     | 0.929 / 0.466       | 0.451 / 0.464     | *0.231* / **0.333** | 0.193 / 0.284     |
> | Fredformer      | 0.340 / 0.338     | 1.588 / 0.981     | 0.898 / 0.495       | 0.396 / 0.447     | 0.261 / 0.378       | 0.194 / 0.285     |
> | iTransformer    | *0.324* / 0.327   | 1.634 / 0.990     | 0.916 / 0.501       | *0.248 / 0.357*   | 0.250 / 0.364       | 0.207 / 0.297     |
> | PatchTST        | 0.338 / 0.335     | 1.650 / 0.998     | 0.893 / 0.502       | 0.426 / 0.464     | 0.246 / 0.361       | 0.198 / 0.286     |
>
>
> > These additional results confirm that while our approach is optimized for long-horizon problems, its underlying principles and training stabilization are broadly effective. This showcases the robust and general-purpose nature of our paradigm. We have included the full results in Table 7 in the appendix of our revised manuscript.
>
> [1] Yue, Wenzhen, et al. "OLinear: A Linear Model for Time Series Forecasting in Orthogonally Transformed Domain." arXiv preprint arXiv:2505.08550 (2025).

---

> ### Author Response · Authors · 2025-11-18
> **Respond to Reviewer kCsT (2)**
>
> ****
> **W2: Theoretical assumptions and  the significance of Theorem 1**
>
> Thank you for your valuable feedback on the theoretical section. We agree that our initial presentation had some points that required clarification and improvement. We have undertaken a significant revision of this section to address your concerns, which we believe also strengthens the paper's overall contribution.
>
> > **Regarding the theoretical assumptions:**
>
> - > **Assumption A3 (Temporal Independence):** As indicated also by Reviewer PbGb. This was a problematic assumption. Upon re-examination, we confirmed that A3 is **redundant** for the derivation of our final error bound. We have therefore removed it as a formal requirement in the revised proof. We now leverage the concept of temporal independence as a key discussion point to formally analyze the structural differences in error accumulation between the AR and DO paradigms. This new analysis, prompted by your and reviewer PbGb's feedback, provides a much stronger and clearer motivation for our BDO approach.
>
> - > **Other Assumptions:** The remaining conditions (e.g., A1, A4) are mild preconditions intended to formalize the problem setting in a way that is consistent with real-world scenarios and standard machine learning practices (e.g., excluding extreme outliers, accounting for measurement noise in labels). They ensure the generality of the bound without imposing unrealistic constraints.
>
> > **Regarding the significance of Proposition 1:**
>
> > The primary purpose of our proposition is not to provide the tightest possible error bound, but rather to serve as a **formal motivation** for rethinking the LTSF design space. We include the answer to this concern in detail in your question 3.
> ****
> **W3: BDO strategy seems trivial,  if it is not accompanied with additional block-level supervision**
>
> >Thank you for this insightful comment. It highlights a crucial aspect of our design philosophy for the BDO paradigm, which we would like to clarify.
>
> >We respectfully argue that the block-level supervision is not an "additional" component but an **integral and inseparable part of the BDO paradigm itself.** The two components are synergistic:
>
> >1. The **recursive structure** creates the architectural framework for generating hierarchical sub-forecasts.
> >2. The **block-level supervision** leverages this structure to enforce a meaningful, causal learning curriculum, which is the mechanism that injects causality into the DO framework.
>
> >One enables the other; together, they form the complete BDO strategy.
>
> >Regarding simplicity, we believe this is a primary strength and a key contribution of our work. Our goal was not to introduce complex new modules, but to propose a fundamental, effective, and extensible paradigm, in line with the principle of Occam's razor.
>
> >To draw an analogy, the concept of a **residual connection** is, in itself, trivial to implement. However, its profound impact on the deep learning community stems from the fundamental principle it enables—allowing for much deeper and more trainable networks. Similarly, we believe BDO is a simple yet powerful paradigm that fundamentally re-frames how a model can learn long-term dependencies in an efficient and causally-aware manner.
> ****
> **W4: The EMA is not novel**
>
> Please see our responses to Q4.
>
> ****
>
> **Q1:** Please see our responses to Weakness 1.

---

> > ### Author Response · Authors · 2025-11-18
> > **Respond to Reviewer kCsT (3)**
> >
> > **Q2: Concerning the theoretical assumptions.** Thank you for this insightful question. It allows us to clarify the motivation behind each of the four conditions in our proposition. Each condition is designed to model a specific factor that influences the final forecasting error, making our theoretical results both general and informative.
> >
> > We will explain them one by one:
> >
> > >**A1 (Bounded Data):** This condition formalizes the practical constraint that the time series values operate within a finite range, $\lambda$. The value of $\lambda$ quantifies the size of the solution space; intuitively, a larger range presents a more difficult prediction task and can lead to larger errors. We believe this is a mild precondition in the context of machine learning for two main reasons:
> >
> >   >1. It serves to exclude extreme outliers, which is a standard data preprocessing step.
> >   >2. It reflects the universal practice of normalizing data before training neural networks.
> >      Therefore, this condition ensures the generality of our bound without imposing unrealistic constraints on the data.
> >
> > >To ensure the rigorious as indicated by the reviewer PbGb. We have improved the statement of this condition by emphasizing that the context is a standard machine learning task and the time interval is finite.
> >
> > >**A2 (Multiple Candidate Forecasts):** This is the central assumption motivating our work. It posits that a single, trained NFM can generate a diverse set of candidate forecasts (e.g., by varying a random state $\gamma$). This is the foundational premise that allows us to apply ideas from ensemble forecasting and forecast combination to the design of a single, end-to-end neural network.
> >
> > >**A3 (Temporal Independence):** This was indeed an error in our initial formulation, and we are grateful for your scrutiny on this point. Upon re-examination, we confirmed that this assumption is **redundant** for deriving the error bound and has been **removed from the formal proof** in our revised manuscript.
> >
> > >However, this insight proved valuable. We now use the concept of temporal independence as a tool in our revised analysis to formally contrast the error accumulation properties of AR, DO, and our BDO paradigms. This has significantly strengthened the theoretical motivation for our work. We sincerely apologize for this oversight and any confusion it may have caused. We are confident the revised analysis is now more rigorous and insightful.
> >
> > >**A4 (Predictive Bias):** This assumption introduces a crucial distinction between the **observed data (X)** and the **true, unobservable data-generating process**. This condition acknowledges a real-world limitation: our training labels (X) are themselves noisy samples, subject to measurement errors and system stochasticity. Therefore, even a perfect model trained on this data will have an irreducible bias (b) relative to the true underlying process. Including this term makes our proposition more rigorous and general by accounting for this fundamental source of uncertainty.
> > ****
> > **Q3: The significance of Theorem/Proposition 1:**
> >
> > Thank you for this question, which allows us to clarify the role and significance of our proposition. Its function is best understood as having two main purposes: one **diagnostic** and one **prescriptive**.
> >
> > > First, the proposition provides an intuitive upper bound that formally decomposes the total forecasting error into its fundamental, and often irreducible, components. It explicitly quantifies how the error is influenced by:
> >
> > - The forecast horizon (T)
> > - The intrinsic bias of the chosen NFM (b)
> > - The stochasticity of the data-generating process ($\sigma$)
> > - The number of candidate forecasts generated (c)
> >
> > > This framework makes it clear *why* forecasting error can never be zero in real-world scenarios and isolates the distinct levers that can be used to mitigate it.
> >
> > > Second, and most importantly, the proposition has a prescriptive function that highlights our core contribution. While much of the recent deep learning literature has focused on reducing model bias (b) through ever-increasing architectural complexity, our proposition formally highlights the **number of forecasts (c)** as a powerful yet underexplored lever for error reduction.
> >
> > > By emphasizing c, our work re-directs attention to the foundational principle of forecast combination. This is the central theoretical motivation for our BDO paradigm, which is designed as a novel and efficient method to implicitly generate and combine multiple forecasts within a single, end-to-end network, a concept largely overlooked in recent SOTA designs.

---

> > > ### Author Response · Authors · 2025-11-18
> > > **Respond to Reviewer kCsT (4)**
> > >
> > > **Q4: How a more powerful NFM should necessarily lead to the reduction of estimation error.**
> > >
> > > >This is a really good question that gets to the heart of our theoretical framework. The proposition shows that a large number of forecasts (c) can reduce the *variance* component of the error, as per the Law of Large Numbers. This naturally raises the question: why then do we need more powerful NFMs?
> > >
> > > >The answer lies in the **predictive bias term (b)**.
> > >
> > > >As our framework makes clear, the total error bound is a function of both the variance (which can be reduced by c) and the model's intrinsic bias b (the gap between the model's expected forecast and the true mean). The strategy of generating multiple forecasts **cannot reduce this bias term.**
> > >
> > > >A more powerful NFM is one that is better able to learn and simulate the true data-generating process. Therefore, a 'more powerful' model, in the context of our proposition, is precisely one that can achieve a **lower predictive bias b**.
> > > ****
> > > **Q5: About Figure.2**
> > >
> > > >Thank you for your question regarding Figure 2. This figure is intended as a conceptual illustration. It aims to visually contrast the causally-structured, sequential representations we expect from an AR-like process with the more monolithic, unstructured representations typical of a standard DO model.
> > >
> > > >While this is a conceptual diagram, we provide concrete empirical evidence to support this intuition in the appendix. Specifically, **the t-SNE visualizations in Figure 12** demonstrate how our BDO paradigm learns the hierarchical and homogeneous representations that we discussed.
> > > ****
> > > **Q6: About the novelty of BDO**
> > >
> > > >Thank you for this insightful question. The novelty of the BDO paradigm is indeed multi-faceted, arising from the synergistic integration of three key components.
> > >
> > > >**1. A Short-to-Long Forecasting Curriculum:** BDO introduces a progressive learning curriculum where the forecast horizon grows with the depth of the network. While the general idea of using shorter forecasts to aid longer ones has early roots (e.g., [Nguyen & Chan, 2004]), we formalize and integrate this concept into a modern, end-to-end deep learning paradigm for LTSF.
> > >
> > > >**2. Synergistic Hierarchical Supervision:** As you correctly identified, the block-level supervision is a key component. Its function is two-fold and integral to the paradigm:
> > >
> > > >* It allows for the full reuse of label information at multiple scales, which amplifies the learning signal (including from auxiliary losses like the frequency loss).
> > > >* It implicitly creates a complex, structured loss function that encourages the learning of hierarchical and causally-aware representations, as we analyze in **Appendix D**.
> > >
> > > >**3. Implicit, Recursive Forecast Combination:** The paradigm recursively reuses its own sub-forecasts by concatenating them with the input. This is not merely a feature-engineering step; it is a mechanism that forces the network to implicitly learn a sophisticated **combination function**, continuously refining its predictions as they pass through the layers.
> > >
> > > >This unique design enables a simple MLP to effectively utilize deeper architectures as shown in our experiment, whereas standard DO models often see performance saturate or degrade with increasing depth. This also distinguishes BDO from other recent methods, including large pre-trained models like Chronos, which operate on entirely different principles.
> > >
> > > >Regarding the ablation studies, we would like to respectfully clarify that we provided comprehensive ablations for the BDO paradigm. The main ablation is in **Section 3.2 (Figure 7)**, and we conduct a more detailed, factor-by-factor breakdown in **Appendix (Table 11)**, which disentangles the effects of the recursive input and the hierarchical supervision. We hope this clarification addresses your concern.

---

> > > > ### Author Response · Authors · 2025-11-18
> > > > **Respond to Reviewer kCsT (5)**
> > > >
> > > > **Q7: About the novelty of applying EMA**
> > > >
> > > > > Thank you for this question. We agree entirely that EMA is a well-established technique, and we have been careful to cite its successful applications in other domains within Section 2.6.
> > > >
> > > > > Our contribution is not the invention of EMA itself, but rather its novel application to identify and solve a critical and largely overlooked problem in the LTSF domain: **severe training instability.** As we motivate with empirical evidence, the non-stationary nature of time series and the chronological train/val/test splits often lead to volatile and inconsistent validation loss curves. This is a fundamental problem for two reasons:
> > > > >
> > > > > - It renders standard **early stopping ineffective**, preventing models from reaching their full potential.
> > > > > - It **compromises the fairness and reliability of model comparisons** across the field. Without a stable training signal, it is difficult to assess whether a new architecture is genuinely superior or if its gains are merely an artifact of a lucky optimization path.
> > > >
> > > > > Therefore, we propose EMA as a simple yet highly effective solution to this specific, domain-relevant problem. To our knowledge, our work is the first to systematically diagnose this instability issue in LTSF and demonstrate that stabilizing the training process is a crucial, foundational step for advancing the state-of-the-art. We believe the novelty and significance of this contribution lie in identifying this critical gap and providing a robust solution for it, thereby establishing a stronger foundation for future research in this area.

---

### Official Review · Reviewer_PbGb · 2025-11-03

**Soundness:** 2
**Presentation:** 1
**Contribution:** 1
**Rating:** 2
**Confidence:** 5

**Summary:**

This paper tackles long-term forecasting by revisiting simple architectures rather than increasing model complexity. The authors argue that well-selected MLPs can match or exceed the performance of recent long-horizon forecasting models. They introduce the Multiple Neural Forecasting Theorem (MNFT) to guide model complexity selection.

However, MNFT provides little novel insight. MNFT essentially restates the law of large numbers under unrealistic independence assumptions for forecast outputs. The theoretical contribution feels forced and disconnected from practical forecasting settings.

**Strengths:**

1. The paper provides valuable empirical evidence that many recent long-horizon models underperform compared to carefully tuned MLPs, reinforcing skepticism toward architectural inflation in this area.

2. The authors attempt to provide a theoretical framework for ensemble-style reasoning about neural forecasting models, which is a valuable direction.

**Weaknesses:**

1. The evaluation relies exclusively on overused long-horizon benchmarks (ETT, Weather, Traffic, Electricity). These datasets are heavily saturated, often exhibit weather-driven dynamics, and are not suitable for assessing general forecasting capability. Accurate weather-related forecasts are known to degrade sharply beyond 2–3 weeks due to chaos in atmospheric systems. To strengthen their claims, the authors should test on broader benchmarks, such as M1, M3, or M4, which are more diverse and less prone to overfitting.

2. The exposition of MNFT is opaque. Figure 1 and its caption are confusing and should be rewritten for clarity.

3. The notation and assumptions in MNFT are inconsistent and often unjustified:

    3.1 Defining “forecasting machines” via tuples is unnecessarily convoluted.

    3.2 Assuming finite means and variances excludes heavy-tailed distributions.

    3.3 Assumption A3 (independent forecast generation) is unrealistic in both direct and multi-step setups.

    3.4 Assumption A4 (bounded expected error leading to bounded forecast error) is tautological and makes the theorem effectively vacuous. Also makes MNFT an LLN-like result. I want to remind the authors that the independence assumptions in the time series domain do not hold.

4. The paper omits key related work, notably NHITS (Neural Hierarchical Interpolation for Time Series), which provides a stronger theoretical connection between network capacity, basis expansions, and approximation guarantees.

**Questions:**

1. Given the large number of available baselines in long-horizon forecasting, why were comparisons limited to DUET, TimeDistill, Timer-XL, iTransformer, TimeMixer, PatchTST, Crossformer, and DLinear? Are these truly the most competitive methods?

2. Typo on line 098 (page 2): the goal should be to approximate Y_f, not X_f.

---

> ### Author Response · Authors · 2025-11-12
> **This is an urgent reply.**
>
> Dear Reviewer PbGb,
>
> We would like to express our sincere apologies for this urgent reply to your critical comments.. First of all, thank you for your thorough review and insightful comments. We understand that your primary reservations concern the theoretical analysis section. Upon reviewing your feedback, we realize that our initial presentation may have led to some misunderstandings that we **are eager to clarify**. We believe that addressing these points will resolve your and other reviewers' major concerns. Please see our point-by-point responses to **the Weakness 3** below.
>
> **1. On the use of tuples in the definition:**
>
> > Our intention in defining the Neural Forecasting Machine via a tuple was to provide a compact and precise notation for the factors influencing the multiple forecasts (e.g., input/output lengths, parameters). This formalism helps to clearly state the subsequent proposition without cumbersome repetition.
> ****
> **2. On Assumption A1 (Bounded/Limited Data Range):**
>
> > This is a mild precondition, whose main purpose is to formalize the exclusion of outliers/anomalies, which are typically handled during data preprocessing. As any real-world dataset is finite, its values are inherently bounded (e.g., simply by its absolute supremum) and we do not add further constrain to the range $\lambda$. Furthermore, normalization, a near-universal step in training neural networks, constrains the data to a limited range. We believe this assumption reflects standard practice without loss of generality.
> ****
> **3. On Assumption A3 (Temporal Independence):**
>
> > We sincerely thank you for this sharp observation. We agree that this assumption is unrealistic and, fortunately, **it is indeed redundant for our derived error bound.** The summation of per-timestep errors to arrive at the final bound in Eq. (17) does not require the errors to be independent at all, and thus does not affect the final result. We will remove A3 as a formal requirement for the proof in our revision.
>
> In addition, we can now leverage this concept explicitly as a **discussion point** to analyze and contrast the error structures of DO, AR, and our BDO paradigms. This discussion, which your comment inspired, significantly strengthens the motivations and advantages for our proposed BDO.
> ****
> **4. On Assumption A4 (Predictive Bias)**
>
> > This distinction was an intentional choice to make our proposition more rigorous and reflective of real-world conditions. We distinguish between the **"true" unobservable process** and the **"observed" label data**, as emphasized in lines 130~134. Assumption A4 acknowledges that our observed/collected data is itself a noisy/inaccurate sample and can be biased due to measurement error. The model's ultimate goal is to predict the true process, and b represents the irreducible gap between the best forecast (of a given NFM) and this true mean. This framing is core to our proposition and clarifies why we treat the observed data not as absolute ground truth, but as one sample from a deeper underlying system. This also answers your question 2, which actually is not a typo.
>
> We are grateful for your detailed feedback, which will help us substantially improve the precision of our manuscript (will be reflected in the revisions). We hope these clarifications will more or less address your major concerns.

---

> > ### Author Response · Authors · 2025-11-13
> >
> > Dear Reviewer PbGb,
> > We are releasing the above explanations first to avoid further misunderstandings, as we found that these major concerns can be resolved immediately.
> >
> > We will address other concerns and questions later.

---

> > > ### Comment · Reviewer_PbGb · 2025-11-17
> > >
> > > **1. Empirical results are limited to long-horizon datasets.***
> > > ```
> > > Beyond the theoretical problems in your BDO framework, I stand by my point that one of the most significant weaknesses of the paper is that it relies solely on the long-horizon datasets to assess the viability of your approach. The long-horizon datasets are as close as it gets to MNIST in the forecasting domain; they have been over-fitted, and it is tough to obtain trustworthy signals from them.
> > > ```
> > >
> > > **3.1. On the use of tuples in the definition:**
> > > ```
> > > Your definition of a forecasting machine can be well translated into the function learning paradigm. In your paper, you are already doing. The tuple machine is not a compact notation.
> > > ```
> > >
> > > **3.2. On Assumption A1 (Bounded/Limited Data Range):**
> > > ```
> > > The boundedness of the data in the time series domain is not a mild assumption. What happens when you have a clear trend in your data? Does your method maintain the theoretical guarantees? Is the presence of trends a common thing in the time series domain? How do you extend your results to prevalent cases like this?
> > > ```
> > >
> > > **3.3. On Assumption A3 (Temporal Independence):**
> > > ```
> > > It is nice that you improved one assumption in your theorem to make it more suited to the time series domain.
> > > Will you provide a revised paper with the improved theorem that does not require the independence assumption?
> > > ```
> > >
> > > **3.4. On Assumption A4 (Predictive Bias):**
> > > ```
> > > The distinction between "unobserved process data" and "observed noisy data" is not clear in the paper.
> > > Are you providing a revised paper with an improved distinction?
> > > ```

---

> > > > ### Author Response · Authors · 2025-11-18
> > > > **Respond to further questions of Reviewer PbGb**
> > > >
> > > > **On the use of tuples in the definition**:
> > > >
> > > > >Thank you for your reminder, we have rewritten the section 2.1. Specifically, we replaced the definition of using tuples with more consice descriptions and function learning notations only.
> > > >
> > > > ****
> > > >
> > > > **On Assumption A1:**
> > > >
> > > > >Thank you for your reminder regarding Assumption A1. We understand your concern and agree that the initial description lacked sufficient context.
> > > > The problem may lie in that our proposition is framed within the context of a standard, data-driven machine learning task. In such a setting, any given time series consists of a finite set of observations. As any finite set of numbers is inherently bounded, the existence of a data range is a practical certainty. Therefore, Assumption A1 reflects this reality rather than imposing a strong theoretical constraint on the underlying process. In this sense, it more resembles a parameter rather than a necessary condition, and so, in our opinion, does not constrain the use of the theoretical result.
> > > >
> > > > >To address this concern, we have revised the manuscript to explicitly state this context. We have added descriptions preceding the proposition to clarify that we are considering a standard machine learning problem operating on finite data. In addition, we now emphasize in our discussion of A1 that because the time series data represent a finite collection of numbers, it is always bounded.
> > > >
> > > > ****
> > > >
> > > > **On Assumption A4 (predictive bias):**
> > > >
> > > > In the original version, we explained this point right after the theorem paragraph. To make it clearer, in the revised version, we have reformulated the notations and explained this distinction in lines 96~98.

---

> > > > > ### Comment · Reviewer_PbGb · 2025-11-18
> > > > > **YOUR CENTRAL PROPOSITION/THEOREM IS INTERNALLY INCONSISTENT AND MISALIGNED WITH THE BDO ALGORITHM**
> > > > >
> > > > > **On Assumption A1:**
> > > > > ```
> > > > > Your argument relies on assuming a specific “data generation process” and imposing a distributional structure on the neural forecasting model. Yet, to derive your bound, you treat the underlying time series as finite solely to enforce an artificial upper bound on its values. This contradicts standard stochastic-process modeling and gives the impression that the assumption is introduced exclusively to make the proof go through. As written, A1 does not describe a stochastic time-series generative process and is incompatible with the behavior of your own model.
> > > > > ```
> > > > >
> > > > > **On Assumption A4**
> > > > > ```
> > > > > You train the forecasting model with MAE, which is equivalent to assuming a Laplace likelihood. Laplace distributions have unbounded support, so imposing a hard bound on prediction errors (A4) is incompatible with the very loss function your method optimizes. In this sense, A4 is even less coherent than A1: the assumption directly contradicts the statistical model implicitly defined by your learning objective.
> > > > > ```

---

> ### Author Response · Authors · 2025-11-18
> **Respond to the remaining concerns of Reviewer PbGb (1)**
>
> Dear Reviewer PbGb,
> We have carefully addressed all the concerns you raised. We agree with your assessment regarding the theoretical framing of our initial proposition. To better reflect its role as a foundational motivation rather than a formal, deeply proven statement, **we have changed the term 'Theorem' to 'Proposition' throughout the manuscript.**
>
> Below, we provide a point-by-point response to your other concerns.
> ****
> **W1: About the empirical evaluation.**
>
> >Thank you for this valuable suggestion. We understand your concern about the benchmark datasets in this sub-area (LTSF) and agree that demonstrating our method's effectiveness on a wider range of datasets is crucial for validating our contributions to the field.
>
> >Regarding the M3/M4 datasets, we found that these specific datasets are less commonly used in recent neural forecasting literature, making it difficult to find established SOTA baselines for a direct, like-for-like comparison.
>
> >Therefore, to address your concern while maintaining a focus on benchmarks relevant to recent neural methods, we have conducted a new set of experiments on several widely-used, diverse datasets in the Kaggle competition, including **Power, Website, NASDAQ, SP500, Carsales, and METR-LA**. For these experiments, we followed the setting from the recent NeurIPS 2025 paper [1], using a 36-step lookback to predict horizons of {24, 36, 48, 60}. To ensure fairness, the performance of competing models is cited directly from their published results.
>
> >The results, presented below, show that our MLP-based model continues to achieve state-of-the-art performance, outperforming recent, more complex architectures even on these new benchmarks. This further underscores the robustness and effectiveness of our proposed paradigm.
>
> **Average Results for Short-Term Forecasting (MSE / MAE)**
> | Model           | CarSales          | Power             | METR-LA             | Website           | SP500               | NASDAQ            |
> | --------------- | ----------------- | ----------------- | ------------------- | ----------------- | ------------------- | ----------------- |
> | **ReNF (Ours)** | **0.289 / 0.306** | **1.335 / 0.885** | **0.793** / *0.414* | **0.233 / 0.348** | **0.225 / 0.333**   | **0.175 / 0.269** |
> | OrthoLienar     | 0.340 / *0.320*   | 1.487 / *0.922*   | 0.838 / **0.402**   | 0.327 / 0.383     | *0.231* / **0.333** | 0.187 / 0.277     |
> | TimeMixer       | 0.338 / 0.336     | 1.484 / 0.937     | 0.881 / 0.499       | 0.415 / 0.456     | 0.241 / 0.353       | *0.186 / 0.281*   |
> | FilterNet       | 0.336 / 0.335     | 1.614 / 0.986     | 0.875 / 0.489       | 0.458 / 0.478     | 0.254 / 0.365       | 0.197 / 0.289     |
> | FITS            | 0.379 / 0.365     | 1.711 / 1.028     | 0.927 / 0.510       | 0.604 / 0.585     | 0.291 / 0.412       | 0.210 / 0.302     |
> | DLinear         | 0.373 / 0.368     | 1.549 / 0.972     | 0.819 / 0.550       | 0.401 / 0.459     | 0.277 / 0.391       | 0.228 / 0.331     |
> | TimeMixer++     | 0.347 / 0.340     | *1.470* / 0.934   | *0.799* / 0.480     | 0.384 / 0.434     | 0.247 / 0.361       | 0.193 / 0.290     |
> | Leddam          | 0.343 / 0.338     | 1.570 / 0.975     | 0.878 / 0.490       | 0.393 / 0.443     | 0.252 / 0.369       | 0.196 / 0.286     |
> | CARD            | 0.355 / 0.338     | 1.547 / 0.942     | 0.929 / 0.466       | 0.451 / 0.464     | *0.231* / **0.333** | 0.193 / 0.284     |
> | Fredformer      | 0.340 / 0.338     | 1.588 / 0.981     | 0.898 / 0.495       | 0.396 / 0.447     | 0.261 / 0.378       | 0.194 / 0.285     |
> | iTransformer    | *0.324* / 0.327   | 1.634 / 0.990     | 0.916 / 0.501       | *0.248 / 0.357*   | 0.250 / 0.364       | 0.207 / 0.297     |
> | PatchTST        | 0.338 / 0.335     | 1.650 / 0.998     | 0.893 / 0.502       | 0.426 / 0.464     | 0.246 / 0.361       | 0.198 / 0.286     |
>
> >We have added these results to a new table in the appendix (Table 7) of our revised manuscript. We hope these extensive new experiments can address your concern.
>
> [1] Yue, Wenzhen, et al. "OLinear: A Linear Model for Time Series Forecasting in Orthogonally Transformed Domain." *arXiv preprint arXiv:2505.08550* (2025).
> ****
> **W2: About the exposition of MNFT and figure. 1**
>
> >Thank you for this feedback. We acknowledge that the original caption for Figure.1 was too brief. In the revised manuscript, we have expanded the caption for Figure 1.
>
> ****
>
> **W4: About the missed related work**
>
> >Thank you for this suggestion. The omission of N-HiTS was an oversight. We have now added it to our Related Work section and used this opportunity to strengthen our discussion of multiple forecasts, which better clarifies our own contributions.

---

> > ### Author Response · Authors · 2025-11-18
> > **Respond to the remaining concerns of Reviewer PbGb (2)**
> >
> > **Q1:**
> >
> > Thank you for this important question regarding our choice of baselines. Our selection principle was to compare against the recent strong state-of-the-art (SOTA) models to provide a rigorous and fair evaluation of our method.
> >
> > > Our baseline set is grounded in the comprehensive benchmark established in, which provides a fair comparison across many models in this area. We are deeply familiar with these benchmarks and selected models that are reported as new top performers.
> >
> > > To ensure our comparison reflects the very latest advancements, we specifically included several models recently published in top-tier conferences, such as **TimeBridge (ICML 2025)** and **Timer-XL (ICLR 2025)**. The chosen baselines (iTransformer, PatchTST, DLinear, etc.) represent a diverse range of the most successful recent architectures, from complex Transformers to simple linear models.
> >
> > > To the best of our knowledge, this curated set represents the current SOTA for these benchmarks under the specified experimental settings. We believe these comparisons robustly validate our claims.
> >
> > ****
> >
> > **Q2:** Please see our response to Weakness 3 and to your further questions.
> >
> > We thank Reviewer PbGb again for her/his constructive feedback. Her/his insightful concerns have led to substantial improvements in both the theoretical framing and the experimental validation of our work. We hope our detailed responses and the corresponding revisions have fully addressed the concerns, and we would be grateful if the reviewer would consider these clarifications in her/his re-evaluation of our paper's contribution.

---

> > ### Comment · Reviewer_PbGb · 2025-11-18
> > **On the poor selection of long-horizon benchmark data.**
> >
> > 1. Even with your “extended” long-horizon datasets, the total sample remains extremely small. SP500 has only 5 series, Nasdaq 12, METR-LA 207, Website 4, Power 2, and Car Sales 10, just 240 series in total. With such a limited sample size, it is not possible to draw realistic conclusions about the performance gains of your method.
> >
> > 2. Several of the chosen forecasting tasks are unrealistic. If one could reliably produce accurate long-horizon forecasts for indices like the SP 500 or Nasdaq, the implications would be financial, not academic. Including such tasks undermines the seriousness of your paper.
> >
> > Please stop perpetuating such a poor selection of benchmark data.

---

> > > ### Comment · Reviewer_PbGb · 2025-11-18
> > > **Updated Score based on the poor quality of theory and empirical studies**
> > >
> > > Given that the theory does not remotely match the algorithm proposed in the paper. Additionally, due to the poor quality of the empirical studies, I will update my review score to hard reject.

---

> ### Author Response · Authors · 2025-11-19
> **Our Final Response to Reviewer PbGb**
>
> We do not understand what makes you so malicious towards this manuscript, and we are astonished by your further responses and actions. We have become fully aware of your hostility and prejudice towards this manuscript and even the entire sub-field.
>
> We are not assuming any specific 'data generation process' or distribution structure at all. All the consideration is about an error bound in a standard machine learning task with a finite number of training data and test data. This criticism is clearly groundless.
>
> Furthermore, we have repeatedly emphasized in the paper that the role of this theorem is merely to serve as a more rigorous motivation for our method. It does not directly participate in the model's design or operation, so the claim that it is 'incompatible with the behavior of our model' is baseless. On the contrary, our experimental results show that the empirical evaluation is consistent with the behavior of this bound.
>
> Lastly, regarding the supplementary experiments, your comments contain numerous errors: for example, the benchmark in our supplementary experiment is about short-term forecasting, not long-term; the numbers you pointed out, '5, 12, 207...', are the number of variates in the datasets, not the 'number of series' for training, evaluating, and testing; all of this data comes directly from Kaggle competitions and has been widely adopted by top-conference papers, not deliberately chosen by us. Please keep professional, read the paper carefully, and take our responses and efforts seriously.
>
> We cannot accept your excessive and groundless accusations against this manuscript. Considering your inconsiderate attitude, this will be our last response to you. Any further replies will be reported directly to the AC.

---

### Note · Authors · 2025-11-20

**Comment:**

We would like to express our gratitude to all the reviewers for their diligent efforts in evaluating this work and for their invaluable suggestions on how to enhance its quality. Following a thorough re-evaluation, the authors acknowledge that the current manuscript contains some inaccuracies and significant scope for refinement. We will carefully consider the reviewers' suggestions and revise the paper to ensure its accuracy and clarity.

**Withdrawal Confirmation:**

I have read and agree with the venue's withdrawal policy on behalf of myself and my co-authors.